

# The combined effect of two westerly jet waveguides on heavy haze in the North China Plain in November and December 2015

Xiadong An[1], Lifang Sheng[1, 2], Qian Liu[3], Chun Li[1, 2], Yang Gao[4], and Jianping Li[5]

[1]Department of Marine Meteorology, College of Oceanic and Atmospheric Sciences, Ocean University of China, Qingdao, 266100, China

[2]Ocean-Atmosphere Interaction and Climate Laboratory, Key Laboratory of Physical Oceanography, Ocean University of China, Qingdao, 266100, China

[3]School of Atmospheric Sciences, and Guangdong Province Key Laboratory for Climate Change and Natural Disaster Studies, Sun Yat-sen University, Guangzhou, 510275, China

[4]College of environmental Science and Engineering, Ocean University of China, Qingdao, 266100, China

[5]Key Laboratory of Physical Oceanography–Institute for Advanced Ocean Studies, Ocean University of China and Qingdao National Laboratory for Marine Science and Technology, Qingdao, 266003, China

*Correspondence to*: Lifang Sheng (shenglf@ouc.edu.cn)

**Abstract.** Severe haze occurred in the North China Plain (NCP) from November to December 2015, with a wide spatial range and long duration. In this paper, the combined effect of two westerly jet waveguides on haze in the NCP was investigated based on visibility observational data and NCEP/NCAR reanalysis data. The results showed that the two Rossby waveguides within the westerly jet originating from the Mediterranean were responsible for haze formation in the NCP. The Rossby wave propagated eastward along the subtropical westerly jet and the polar front jet, causing an anomalous anticyclone over the Sea of Japan and anticyclonic wind speed shear at 850 hPa over the NCP, which enhanced the anomalous descending air motion in the middle and lower troposphere and subsequently resulted in a stable atmosphere. Furthermore, the Rossby wave weakened the East Asia trough and Ural ridge, and strengthened the anomalous southerly wind at 850 hPa over the coastal areas of east China, decelerating the East Asia winter monsoon. The above meteorological conditions modulated haze accumulation in November and December 2015. Meanwhile, continuous rainfall related to ascending motion due to Rossby wave propagation along the subtropical westerly jet occurred in a large area of southern China. The latent heat released by rainfall acted as a heat source, inducing convection over South China. This further strengthened the ascending motion over South China so that the descending motion over the NCP was maintained, favoring the maintenance of severe haze. This study is of great significance

to elucidate the formation and maintenance mechanism of large-scale haze in the NCP in late fall and boreal winter.

## 1 Introduction

Haze is the phenomenon of reduced visibility caused by the increase in aerosols or the hygroscopic growth of aerosols at a

high relative humidity (Ma et al., 2014). In December 2015, the air quality was poor in the North China Plain (NCP), especially

in the Beijing-Tianjin-Hebei region (Chang et al., 2016; Zhang et al., 2016; Zhang et al., 2019). Based on the emissions level

in December 2015 and the Weather Research and Forecasting-Community Multiscale Air Quality (WRF-CMAQ) regional

model, Zhang et al. (2019) found higher monthly mean PM2.5 concentrations. The causes of haze in China, except for pollutant

emissions, the weather conditions and climate change also play a significant role in modulating haze formation, distribution,

maintenance and change (Ding et al., 2009; Tai et al., 2012; Zhang et al., 2013). Yang et al. (2016) found that changes in

meteorological parameters contributed 17 (±14)% to the increasing trend in $PM_{2.5}$ concentration from 1985 to 2005. Dang

and Liao (2019) found that severe winter haze days showed large interannual variations in frequency and intensity, which were

mainly driven by changes in meteorology.

Winter haze- fog in northern China in recent years are related to weak winter circulation (Wu et al., 2014). Some studies have

noted that the circulation in mid- and high latitude regions plays an important role in wind variations in northern China (Wang

et al., 2016; Li et al., 2017; He et al., 2019). Zhao et al. (2013) and Zhang et al. (2013) showed that the weak East Asian winter

monsoon (EAWM) is particularly unfavorable to the outward transport of aerosols in northern China, which results in

continuous strong haze in northern China. In addition, the stable weather conditions associated with the Siberian high pressure

are also conducive to the accumulation and maintenance of haze (Zhang et al., 2016; Liu et al., 2017). Li et al. (2019) reported

that the Eurasian teleconnection (EU) at 500 hPa is the most important pattern affecting haze- fog in northern China, and its

contribution rate is 45%. In the negative EU, there is a positive anomaly in Europe and East Asia and, a negative anomaly in

Siberia at 500 hPa. As a result, the cold air mass from the arctic is too weak to reach northern China (Ren et al., 2010; Zhai et

al., 2016), which is not conducive to the dissipation of haze.

According to many studies, the North African-Asian (NAA) jet has an important impact on the Asian climate in winter (Syed

et al., 2006; Feldstein and Dayan, 2008; Wen et al., 2009; Ni et al., 2010; Li et al., 2013). On the decadal timescale, the strength

of the jet is highly consistent with the frequency of cold air temperature extremes in China (Chen et al., 2013). The divergence

in the upper troposphere induced by the subtropical westerly jet waveguide is important for spatially large and persistent winter

rainfall in South China (Li and Sun, 2015; Ding and Li, 2017). Based on previous theoretical work, Xu et al. (2019) noted that

the British-Baikal Corridor teleconnection on the polar front jet waveguide in Eurasia has an important impact on the climate

of East Asia in summer. Zhang et al. (2019) found that the winter concurrent meridional shift in the East Asian jet streams is

related to the east-ward propagating Rossby wave. Huang et al. (2019) further found that the concurrent change in the location

of the subtropical and polar front jet in East Asia in winter has an important impact on the climate of East Asia. In addition,

previous studies showed that as the teleconnection wave source location, the anomaly in atmosphere circulation related to sea

surface temperature (SST) in the North Atlantic can significantly influence winter weather in the Northern Hemisphere

(Wallace and Gutzler, 1981; Peng et al., 1995; Czaja and Frankignoul, 2002; Li and Betas, 2007), which can influence haze in

China (Xiao et al., 2014; Gao and Chen, 2017). Chen et al. (2013) noted that the cold air activity was possibly associated with

the development of the Siberian high pressure and a wave train from the North Atlantic. Yang et al. (2019) stated that the East

Asian wind and temperature anomalies during boreal winter are determined by the combination of the two wave trains

propagating along the subtropical jet and the polar front jet. The SST anomalies in the North Atlantic can induce a downstream

Rossby wave train, resulting in anomalous circulations over the NCP, which are favorable for haze over central and south

China (Feng et al., 2019; Wang et al., 2019).

In summary, the Rossby waveguide within the East Asian upper westerly jet has important influences on the East Asian climate.

Previous research on the meteorological parameters influencing haze in the NCP has focused mainly on local meteorological

conditions in the middle and lower troposphere, or only the correlation between haze in China and North Atlantic SST or other

patterns is analyzed only, while studies about the specific mechanisms behind the influence of large atmospheric circulations

in the upper troposphere on haze, especially the combined effect of two westerly jet waveguides, are rare. Given the above

content, the objective of the present study is to determine whether the effects of two westerly jet waveguides on haze are

significant in the NCP in November and December 2015 and, if so, to identify the principal mechanism behind the effects of

two westerly jet waveguides on haze.

The structure of this paper is as follows: data and methodology used in this paper are described briefly in Section 2. Section 3

presents the haze event and describes major meteorological parameters and atmosphere circulation patterns during haze events.

Section 4 demonstrates the influencing mechanisms of two westerly jet waveguides on haze events. Further discussions and

conclusions of our findings are provided in Section 5.

## 2 Data and Methodology

### 2.1 Data

Visibility station data from the stations in the region (15-55°N, 105-135°E) in November and December 2015 were from the

China Meteorological Administration (CMA). These data were recorded four times daily (02:00, 08:00, 14:00, and 20:00, local

time (LCT)). The data were averaged monthly in this study.

The monthly mean and daily mean reanalysis data are from the National Centers for Environmental Prediction-National Center

for Atmospheric Research (NCEP/NCAR) NCEP-DOE AMIP-II Reanalysis (R-2) dataset (Kanamitsu et al., 2002). The dataset

covers a 39-yr period from 1979 to 2017, with a latitude-longitude spatial resolution of 2.5º × 2.5º, including the geopotential

height, wind vector, relative humidity, air temperature, and omega data at each standard level from 10 hPa to 1000 hPa. The

daily precipitation data from November to December 2015 are provided by the NASA/Goddard Space Flight Center (Huffman

et al., 2014).

### 2.2 Methodology

Haze was defined as a day when the daily mean visibility and relative humidity were less than 10 km and 80%, respectively,

and no rain, snow, sand and dust storms occurred in accordance with the standards set by the CMA (China Meteorological

Administration, 2010).

Visibility station data were interpolated on the regular grid of 0.5°× 0.5° using Cressman interpolation method that was

proposed by Cressman in 1959. Cressman objective analysis adopts the method of successive corrections, which has been

widely used in various climate diagnosis analysis and numerical simulation studies. The most important Cressman objective


analysis is the determination of the weight function $W_{i,j}$ (Feng et al., 2004) (1-2):

$$W_{i,j} = \frac{R^2 - r_{i,j}^2}{R^2 + r_{i,j}^2} \qquad r_{i,j}^2 < R, \tag{1}$$

$$W_{i,j} = 0 \qquad r_{i,j}^2 > R, \tag{2}$$

Where R is the influence radius, $r_{i,j}^2$ is the distance between two interpolation points, i and j are each interpolation point. R

are 5, 4 ,3 in this paper. Unit of R is degree.

$$\emptyset_{i,j} = \sum_{k=1}^{N} W_{i,j}^k \emptyset_{obs}^k \Big/ \sum_{k=1}^{N} W_{i,j}^k, \tag{3}$$

Where $\emptyset_{i,j}$ is the function of interpolation and $\emptyset_{obs}^k$ is station data. N is number of stations.

To analyze the anomalous propagation of Rossby waves, we calculated horizontal stationary wave activity flux to show the

propagation of wave energy using the method of Plum (1985) (4):

$$F = p_0 cos\emptyset \begin{pmatrix} \frac{1}{2a^2 cos^2\emptyset} \left[ (\frac{\partial \psi'}{\partial \lambda})^2 - \psi' \frac{\partial^2 \psi'}{\partial \lambda^2} \right] \\ \frac{1}{2a^2 cos^2\emptyset} \left( \frac{\partial \psi'}{\partial \lambda} \frac{\partial \psi'}{\partial \phi} \right) - \psi' \frac{\partial^2 \psi'}{\partial \lambda \partial \phi} \end{pmatrix}, \tag{4}$$

Here, $F$ is the horizontal stationary wave activity flux, $p_0$ is pressure / (1000 hPa), $\psi'$ is the perturbation stream function, $a$ is

Earth's radius, $\emptyset$ is the latitude, and λ is the longitude.

The East Asian winter monsoon composite index defined by He and Zhou (2012) was used to characterize the East Asia winter

monsoon. The formula is as follows (5-8):

$$I_1 = Norm[\bar{p}_s(40°{\sim}60°N, 80°{\sim}125°E)] , \tag{5}$$

$$I_4^* = -1 \times I_4 = -1 \times Norm[\bar{h}_{500}(25°{\sim}45°N, 110°{\sim}145°E)] , \tag{6}$$

$$I_5 = Norm[\bar{u}_{300}(25°{\sim}40°N, 80°{\sim}180°E) - \bar{u}_{300}(45°{\sim}60°N, 60°{\sim}160°E)] , \tag{7}$$

$$EAWMII = \frac{I_1 + I_4^* + I_5}{3}, \tag{8}$$

Here, Norm represents standardization, and $\bar{p}_s$, $\bar{h}_{500}$, and $\bar{u}_{300}$ are the mean sea level pressure, 500 hPa mean geopotential

height and 300 hPa mean zonal wind over the region defined above, respectively.

Please note that all the anomalies calculated in this paper were based on the climatological mean during the thirty-year period

of 1981–2010. For example, the calculation procedure can be written as (9)

$$hgt\_a_{Nov2015} = hgt_{Nov2015} - hgt\_m_{Nov \ during \ 1981 \ to \ 2010}, \tag{9}$$



Here, $hgt\_a_{Nov2015}$, $hgt_{Nov2015}$ and $hgt\_m_{Nov\ during\ 1981\ to\ 2010}$ represent the geopotential height anomaly in November

2015, the geopotential height in November 2015 and geopotential height mean in November 1981 to 2010.

## 3 Persistent haze events and associated weather patterns

Heavy haze occurred over the NCP in November and December 2015. The locations where the average monthly visibility was

less than 10 km covered northern China and eastern China (Figure 1). The haze events were characterized by wide range,

strong intensity and long duration, which was consistent with the analysis of Chang et al. (2016), Zhang et al. (2016) and

Zhang et al. (2019). There were 22 haze days, accounting for more than 70% of the total, in both November and December

2015 (Figure 2). In November, the mean monthly visibility over the NCP was 8.33 km with a minimal value of 4.14 km. A

regional mean visibility less 10 km appeared on 4-16, 18-22 and 27-30, with minimal daily values of 5.16 km, 5.86 km and

4.14 km, respectively. Similarly, the mean monthly visibility in December over the NCP was 8.63 km with a minimal daily

value of 2.62 km. The dates when the regional mean visibility was less than 10 km were 6-14, 19-24 and 28-31 December,

with minimal daily values of 3.87 km, 2.62 km and 6.00 km, respectively.

Static stability is expressed by the vertical difference in the temperature between 1000 hPa and 850 hPa (Liu et al., 2017),

which is on average 4.5°C-7.5°C in the NCP. The value of the static stability was small so that the atmosphere was relatively

stable (Figure 3a). In addition, the vertical profiles of the temperature anomalies between 1000 hPa and 700 hPa show warm

anomalies in the middle and lower troposphere and cold anomalies near the surface (Figure 3e), which further indicates that

the atmosphere is relatively stable. When haze occurred, the mean monthly relative humidity (RH) in the NCP was less than

70% (Figure 3b), and the regional daily relative humidity was also relatively low (not shown). According to the haze

identification conditions issued by the China Meteorological Administration (2010), haze rather than haze- fog can be

considered the main pollution event in November and December 2015. An RH of approximately 55-70% may promote the

liquid phase reaction of aerosol particles. RH has an important influence on the hygroscopic growth of aerosol particles (Ma

et al., 2014), which can promote the photochemical reaction of aerosol particles. Wu et al. (2019) found that the air over the

NCP during the haze episode was humid, with an average simulated RH of approximately 71%.

The ground was mainly controlled by an anticyclone in November and December 2015 according to the ground wind field at 10 m. The NCP was in the center of the anticyclone, and the wind speed was very weak (Figure 3c). However, in the periphery

of the NCP, the wind speed was relatively strong (Figure 3c). The horizontal wind speed was a negative anomaly from the lower troposphere to the upper troposphere (Figure 3e). The 850 hPa wind field showed a northeast-southwest wind speed shear (Figure 3d), which corresponded to negative vorticity anomalies. The atmosphere was mainly composed of descending movement, so the stratification was relatively stable. According to Liu et al. (2017), the frequency of haze occurrence was 24.5% under low wind speed conditions at the ground and low stability over 1303 noncold wave days, accounting for the

highest proportion of the four conditions (including low wind speed and low stability (20.4%), high wind speed and low stability (14.4%), and high wind speed and high stability (14.6%)). In addition, compared with surface temperature, surface specific humidity, cloud fraction, precipitation rate, and upward flux (UP), the contributions of horizontal wind and the boundary layer height to $PM_{2.5}$ concentration in winter in eastern China were 37% and 25%, respectively (Yang et al., 2016). Therefore, wind speed and atmospheric stability play vital roles in haze formation during November and December in the NCP.

From the 850 hPa wind vector, it can be found that there was an anticyclone circulation in southern China. The southerly wind on the northwest side of the anticyclone circulation prevented the southward dispersion of pollutants over the NCP. At the same time, the anticyclone circulation weakened the EAWM in the NCP. According to the winter monsoon composite index defined by He et al. (2012), the EAWM index in November and December 2015 was -0.29 (not shown). The weak EAWM was beneficial to the occurrence and maintenance of haze.

Figure 4 shows the mean monthly circulation anomalies and the average jet positions in November and December 2015. At 200 hPa (Figure 4a), there was a clear northerly and southerly anomaly in the meridional wind from west to east within the mid-high latitude westerly jet. The north-south anomaly in the meridional wind corresponded to the geopotential height anomaly, similar to the wave train. This was characterized by a Rossby wave corresponding to the teleconnection patterns and stationary Rossby waves in northern hemisphere winter, as proposed by Hoskins et al. (1993) and Hsu et al. (1992). This was

also similar to the teleconnection of the Northern Hemisphere Silk Road in summer proposed by Lu et al. (2002). There was a strong northerly anomaly over the European continent. The strong northerly wind formed a convergence of cold air in the Middle East and the Mediterranean Sea, which intruded into the entrance of the westerly jet stream and propagated eastward





in the form of a wave train along the westerly jet stream. The centers of the southerly wind anomaly were located over the Arabian Peninsula and East Asia, and the centers of the northerly wind anomaly were located over the European continent and Central Asia. The northerly and southerly wind anomalies over the Asian continent resulted in anomalous circulation southern China, with abnormal ascending movement on the eastern side of the cyclonic anomaly according to the value of divergence at 850 hPa and 200 hPa and the omega value at 500 hPa. The NCP experienced an anticyclonic anomaly with abnormal descent. Atmospheric motion ascended over southern China and descended over the NCP to form a north-south circulation system from 20°N to 50°N at 112°E-120°E (Figure 4a, Figure 6). In addition, the NCP was located in the middle of the cyclonic and anticyclonic anomalies, with strong southerly winds at 850 hPa (Figure 4b, Figure 4c), weakening the EAWM and favoring haze accumulation. There was a negative phase of the EU at 500 hPa (Figure 4b): the negative geopotential height anomaly over Siberia weakened the Ural ridge, and the positive geopotential height anomaly over the Sea of Japan weakened the East Asian trough. According to Li et al. (2019), the direct effect of the EU negative phase is that the arctic cold air does not easily flow to East Asia, and the EAWM weakens. Thus, a negative EU may have an important impact on haze. Due to the anticyclonic anomaly in the troposphere over the Sea of Japan, the descending movement over the NCP stabilizes the atmosphere, conductive to the haze accumulation. In southern China, there was mainly ascending motion, with an anomalous southwesterly wind at 850 hPa (Figure 4c). The convergence of southwesterly airflow was conducive to the occurrence of large-scale continuous precipitation over the southern China, as reported by Li and Sun (2015) and Ding and Li (2017). As shown in Figure 2, strong precipitation occurred in November and December 2015 in southern China (20°N-30°N, 100°E-120°E), i.e., during November 6-13, 14-17, and 18-21 and December 2-6, 7-10 and 21-23. The north-south circulation systems were consistent, which was implicative of a barotropic stable circulation structure triggering long-lasting of the haze events.

To more clearly depict the circulation in detail during the persistent haze events, the circulation field was composited by selecting haze events lasting at least three days with a mean daily visibility of less than 8 km (Figure 5). The periods are November 9-15, 19-21 and December 6-10, 19–25. The results during the severe haze events are in generally consistent with those of the monthly average. The locations of the centers of convergence and divergence at 850 hPa and 200 hPa and the anomalous meridional wind at the 200 hPa are similar to the monthly average locations (Figure 4a, 4c and Figure 5a, 5c), but the intensity of the anomalous meridional wind over the Mediterranean and East Asia seems to be stronger, and the strength of



the divergence is enhanced as well. There is a relatively strong trough over the North Atlantic, a weak trough over the Ural

Mountains and a trough over the North Pacific at 500 hPa. In addition, there was a weak Southern Branch trough over the

southern Qinghai-Tibet Plateau and southern China. The subtropical high over the western Pacific was stronger and located in

the west (Figure 5b). This circulation situation reduced the amount of cold air intruding China and the winter monsoon also

weakened. The Southern Branch trough provides rising conditions for precipitation over southern China. The southerly airflow

on the west side of the subtropical high (Figure 5c; near 115°E) carried a large amount of water vapor originating from the

South China Sea into southern China. In contrast, convergence of the southerly wind occurred over the NCP, leading to haze

formation.

Through the analysis above, we found that the main meteorological factors modulating the large-scale persistent haze in the

NCP are wind vector and atmospheric stability. The anomalous southerly wind over the eastern coast of China weakens the

winter monsoon, and the anticyclonic shear of the wind at 850 hPa in the NCP yields an increase in negative vorticity and the

subsequent enhancement of atmospheric descent. In addition, the anomalous anticyclone over the Sea of Japan strengthens the

descending air motion and increases atmospheric stability. From the geopotential height anomaly at 200 hPa (not shown), 500

hPa (Figure 4b), 850 hPa (Figure 4c) and the tropospheric vertical motion (Figure 6), it can be seen that the upper and lower

levels exhibit barotropic structures. The north-south anomaly of the meridional wind from west to east appears in the upper

tropospheric subtropical westerly jet, which is similar to the Silk Road teleconnection proposed by Lu et al. (2002). The

anticyclone anomaly over the NCP strengthens the descending motion, and the ascending motion related to upper divergence

and lower convergence over southern China is strong. The north-south circulation system leads to the maintenance of haze in

the NCP, whereas large-scale precipitation occurred over southern China.

**4 Evolution of two westerly jet waveguides and principal mechanism analysis**

To explore the influence mechanism of Rossby wave in subtropical westerly jet on haze, the first leading EOF mode of the

200 hPa zonal and meridional winds from November 1979 to December 2017 was calculated. The first mode of zonal wind

represents the north-south movement of the subtropical jet (Figure 7a), following Hong et al. (2016). This mode's variance is



25.3%. From the first modal time series, it can be seen that the exponent is positive in 2015, meaning that the jet is strong. The first mode of the meridional wind is a Rossby waveguide manifested by the north-south anomaly of the 200 hPa meridional wind, following Li et al. (2017). The variance is 23.4%. The first modal time series shows that the exponent has a negative index in 2015, with strong intensity (Figure 7c). The northerly wind anomaly appears in the Mediterranean region and the

Indian Peninsula region, and the southerly wind anomaly appears in the Arabian Peninsula region, North China and the Yellow Sea.

To investigate the relationship between the position change in the jet and the Rossby waveguide, the time series of the first mode of zonal wind and the meridional wind EOF were used to regress the 200 hPa wind field and meridional wind, respectively (Figure 8). The results of the regression are very similar as shown in Figure 8a and Figure 8b. That is, the position

movement of the jet and Rossby waveguide was closely related in winter, which is basically consistent with the results of Hong et al. (2016) on the relationship between the Silk Road teleconnection (summer) and the change in jet stream location.

To reveal the influence of the jet waveguide on atmospheric circulation in the middle and lower troposphere, 500 hPa geopotential height anomalies and 850 hPa wind field anomalies were regressed by the time series of the first EOF mode of meridional wind (Figure 9). The EU negative phase appears in the regressed geopotential height field. The two positive centers

of the regression coefficients of the geopotential height anomaly are located over the Mediterranean Sea and the adjacent Sea of Japan areas. The positive geopotential height anomaly near the Sea of Japan weakens the East Asian trough, weakening the EAWM. In addition, the anticyclone anomaly also causes the southerly wind anomaly at 850 hPa over the eastern coast of China, further weakening the winter monsoon. The center of the negative geopotential height anomaly is located in Siberia, weakening the Ural ridge and weakening the Lower Siberian High and favoring haze formation. The regression coefficient of

the wind field anomaly shows that there is a southerly anomaly over the NCP, and a southwesterly anomaly in South China. The southerly anomaly over the NCP tends to weaken the winter wind, while the southwesterly anomaly over South China carries water vapor from the Bay of Bengal and South China Sea to South China, providing favorable moisture conditions for precipitation in South China.

To further examine the mechanism behind the propagation of the Rossby waveguide, two-dimensional wave activity flux was

calculated and generated according to the formula for the wave action flux defined by Plum (1985). Figure 10 shows that the

European continent is the source of the Rossby wave. Rossby waves propagate into two pathways: a subtropical westerly jet and polar front jet. The Rossby wave in the polar front jet was stronger than that in the subtropical westerly jet. Rossby waves in the polar front jet, named the EU, resulted in energy dispersion over the Siberian area, leading to a negative geopotential height anomaly in this location, weakening the Ural ridge and preventing cold air advection towards China. The remaining

energy from the easterly propagation caused the positive geopotential height anomaly over the Sea of Japan, weakening the East Asian trough and inducing the southerly wind anomaly over the eastern coast of China. The southern branch of the Rossby wave propagated within the subtropical westerly jet that passed over the Arabian Peninsula, India and other places. Almost all of its energy was dispersed within South China, resulting in a large amount of heavy rainfall in South China. The latent heat released by precipitation as a heat source strengthened the abnormally ascending motion in South China, which further

strengthened the anticyclone anomaly in the NCP. The wave action flux was obviously enhanced in the Sea of Japan area, which may be due to the concurrent effect of two waveguides superposing in this area. Thus, the energy from the southern branch of the Rossby wave is very weak over the Sea of Japan, but the anticyclone anomaly over the Sea of Japan is still very strong.

Based on the analysis above, a diagram of the Rossby waveguide of the westerly jet affecting haze events in the NCP is drawn

(Figure 11, Figure 12). The Rossby wave originated from the Mediterranean region (Li and Sun, 2015) and propagated eastward along the westerly jet, resulting in an anomalous anticyclone over the Sea of Japan and strengthening descending air motion over the NCP. In addition, there was a southerly wind anomaly at 850 hPa over the eastern coastal areas of China, which weakened the winter monsoon (Figure 12). Both of these features are conducive to haze accumulation. In addition, the Rossby waveguide also caused a strong abnormal upward movement over South China, which combined with the rich water vapor

carried by the southwesterly airflow resulted in large-scale continuous rainfall (Figure 11, Figure 12). The latent heat released by precipitation as a heat source strengthened the abnormal ascending motion over South China, which further helped to maintain the sinking motion in the NCP. A local circulation in the southern and northern parts of China was formed (Figure 12): the air rises in the south and subsides in the north, facilitating the intensification of the descending air motion in the NCP and the persistence of haze.

## 5 Discussions and conclusions

A wide range of haze pollution occurred in the NCP in November and December 2015. The meteorological factors, climatic system and mechanisms affecting haze were elucidated in this study. Haze was mainly modulated by the weakening of the EAWM and the strengthening of descending air in the NCP. The Rossby waveguide in the westerly jet originating from the Mediterranean region was the main mode fostering the atmospheric circulation patterns tightly associated with haze formation in the NCP.

The Rossby wave propagated easterly primarily along two pathways. The northern path seemed to yield positive geopotential height anomalies in the troposphere over the Sea of Japan and the NCP, strengthening the descending air movement over the NCP. In addition, it weakened the East Asia trough and Ural ridge, resulting in the southerly wind anomaly at 850 hPa over the eastern coast of China, weakening the winter monsoon, and leading to subsequent haze accumulation. Due to the influence of the southern path of Rossby wave, there was abnormal upward air movement over South China, which combined with a large amount of water vapor carried by the 850 hPa anticyclone circulation over South China, provided favorable conditions for continuous rainfall in South China. As a heat source, the latent heat of condensation released by precipitation, further strengthened abnormal upward movement in South China, favorable for the subsidence and haze formation in the NCP.

The two westerly jet waveguides may be the main mechanism leading to the occurrence and maintenance of large-scale haze in the NCP in November and December 2015. The linear regression coefficient of PC1 (bar) of the leading EOF mode for the 200 hPa meridional wind anomaly onto the mean visibility anomaly in November and December is approximately 0.38 from 1980 to 2015 (Figure 13). The location of the EU mentioned above is slightly different from that of the EU defined by predecessors. To describe the EU more accurately in this study, a new EU index (EUI) was defined according to the definition of the EUI of Wallace and Gutzler (1981).

$$EUI = -1/4 \times hgt500\_ano \, (40°N, 0°) + 1/2 \times hgt500\_ano \, (50°N, 50°E) - 1/4 \times hgt500\_ano \, (40°N, 140°E)$$

Here, hgt500_ano is the mean geopotential height anomaly at 500 hPa in November and December.

The linear correlation coefficient of the PC1 of the leading EOF mode for the 200 hPa meridional wind anomaly and EUI is approximately 0.92, which further shows that the combined effect of the two waveguides may have an important influence on

heavy haze in the NCP. In addition to 2015, the two waveguides also existed in 1989, 1994, 1996, 2004, 2006, and 2011. The

visibility anomaly in the NCP in these years was negative. However, the two westerly jet waveguides were also strong in 1982,

1986, 1988, 1991, and 1992, and the visibility anomaly in the NCP was positive (Figure 13). In addition, compared to the

waveguides in 2015, the two westerly jet waveguides were weak or out of phase in 2000, 2002, 2003, 2007, 2013, and 2014,

while the visibility anomaly in the NCP was negative. This means that under similar pollutant emission conditions, the large-

scale circulation conditions that cause air pollution may take on diversified modes. This paper mainly focused the haze in

November and December 2015; therefore, the robustness of the mechanism may be further verified if other haze events can

be evaluated in future studies. We can understand the complexity behind the climatic causes of severe air pollution from some

existing studies. For example, the combined effects of the negative North Atlantic Oscillation (NAO) and El Niño worsened

air conditions over central and south China in the winters of 2000 and 2003 (Feng et al., 2019). The autumnal increased number

of haze days in 2013 and 2014 was closely associated with simultaneous sea surface warming in the western North Pacific

sector and the North Atlantic subtropical sector (Wang et al., 2019).

**Acknowledgements**

This study was supported by the National Natural Science Foundation of China (Grant No. 41675146) and Fundamental

Research Funds for the Central Universities (Grant No. 201941006). All authors would like to express their great thanks to the

China Meteorological Administration (http://data.cma.cn/), NCEP/NCAR (https://www.esrl.noaa.gov/psd/data/gridded/), and

NASA (https://pmm.nasa.gov/data-access/downloads/gpm) for supplying the research data. They are also very grateful for

anonymous reviewers for their kind efforts that improved the quality of this manuscript significantly.

**Data availability**

The visibility observational data is available at the China Meteorological Administration (http://data.cma.cn/). The reanalysis

dataset is available at NCEP/NCAR (https://www.esrl.noaa.gov/psd/data/gridded/). The precipitation data is available at NASA

(https://pmm.nasa.gov/data-access/downloads/gpm).

**Author contribution**

XA, LS, CL and JL designed the study. LS and YG acquisitioned the funding. LS obtained observation data. XA and QL dealt with the visibility observational data. XA downloaded, analyzed the reanalysis data and prepared all figures. XA led the writing

with the help of LS, QL and YG. All the authors discussed the results and commented on the paper.

**Competing interests**

The authors declare no conflicts of interest.

**Financial support**

This study was supported by the National Natural Science Foundation of China (Grant No. 41675146) and Fundamental

Research Funds for the Central Universities (Grant No. 201941006).

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

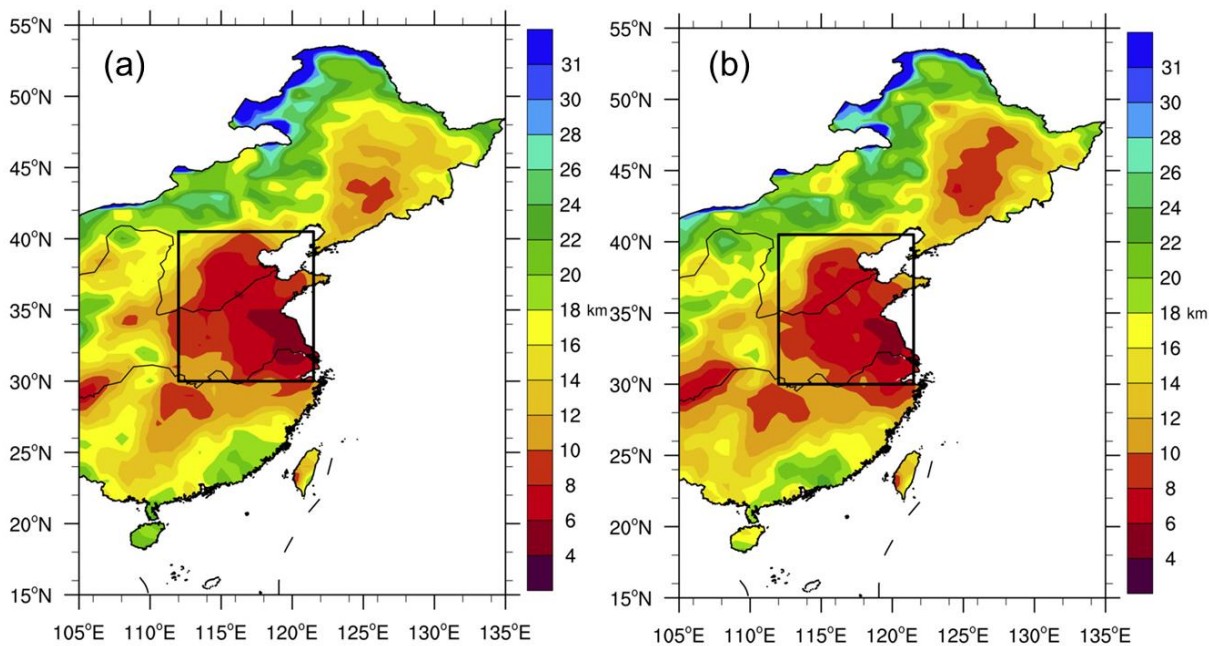

**Figure 1: Spatial distribution of monthly mean visibility (unit: km, shading) in (a) November and (b) December 2015. The black box indicates the NCP (30°N-40.5°N, 112°E-121.5°E). Shading indicates the value of visibility.**





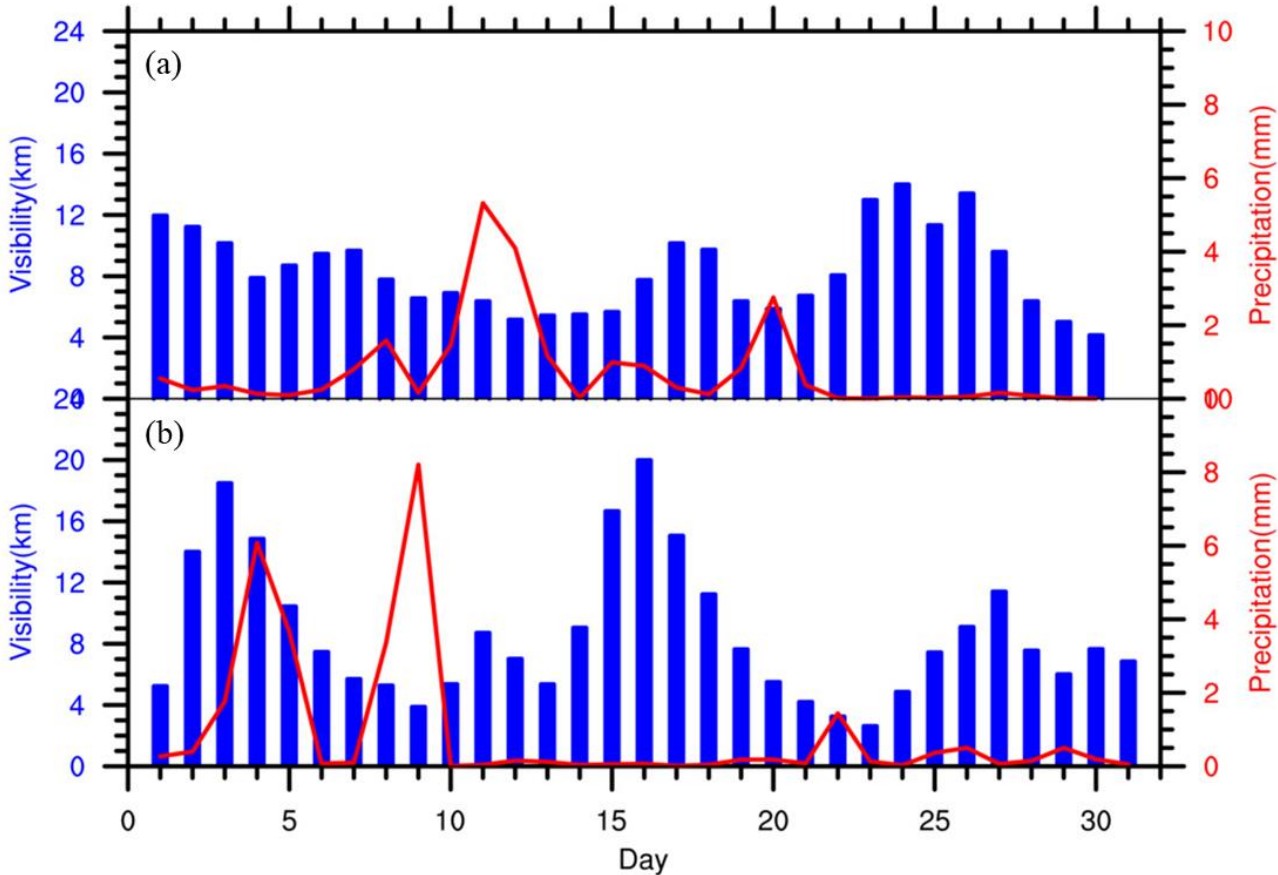


**Figure 2: Variations in the regional mean daily visibility (bars; unit: km) is obtained by calculating the average value of all stations data in the NCP (30°N-40.5°N, 112°E-121.5°E) and the regional mean daily precipitation (lines; unit: mm day$^{-1}$) in southern China (20°N-30°N, 100°E-120°E) (a) in November and (b) in December 2015.**

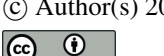



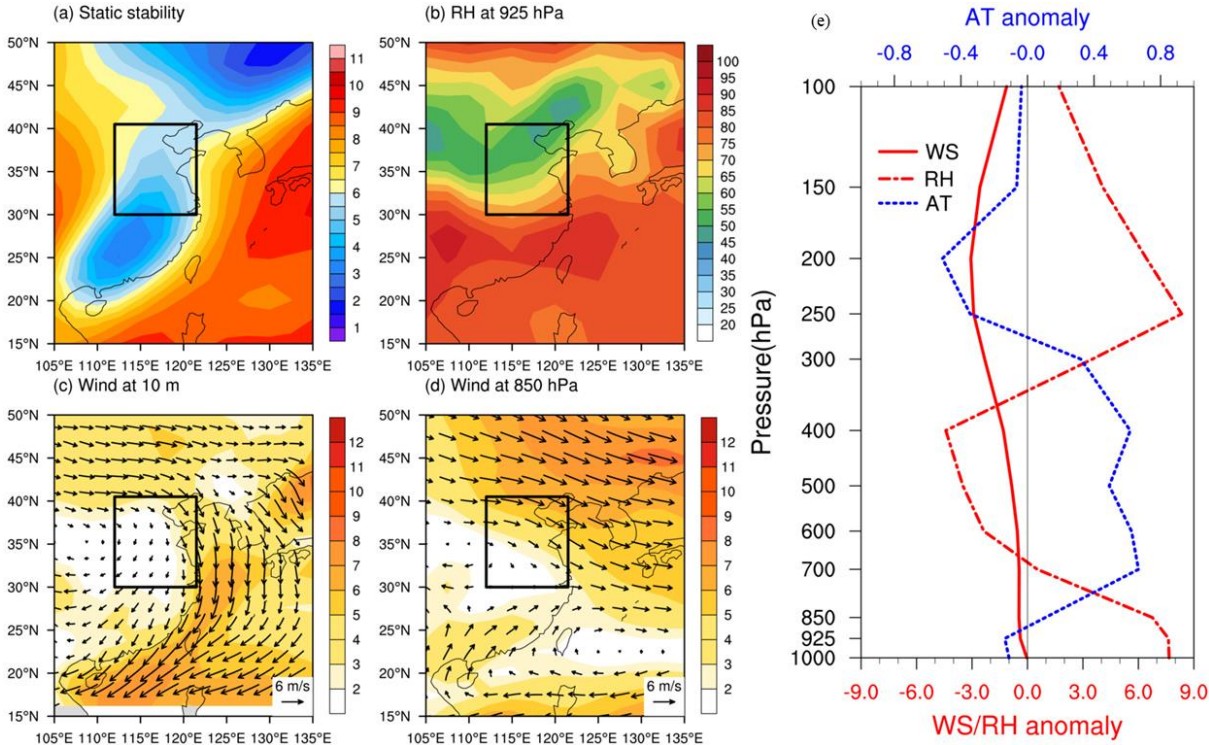

**Figure 3: Average spatial distribution of the static stability (unit: °C; Figure 3a); relative humidity at 925 hPa (unit: %; Figure 3b); wind vector (unit: m s$^{-1}$) at 10 m (Figure 3c) and 850hPa (Figure 3d) in November and December 2015, with shading indicating the wind speed in the appropriate level; anomalous vertical distribution of air temperature (AT) (blue dashed line), horizontal wind speed (WS) (red solid line) and relative humidity (RH) (red dash-dotted line) in NCP in November and December 2015 (Figure 3e).**






**Figure 4: Anomalous weather and westerly jet maps in November and December 2015: (a) meridional wind (black contours, CI (contour interval) = 1 m s⁻¹, solid (dashed) lines represent southerly (northerly) wind anomaly), divergence (shading, $10^{-5}$ m s⁻¹) and westerly jet (green line) (zonal mean wind, CI > 30 m s⁻¹) at 200 hPa; (b) geopotential height (black contours, CI = 10 gpm (geopotential meters), solid (dashed) lines represent positive (negative) geopotential height anomaly) and vertical velocity (shading, $10^{-2}$ Pa s⁻¹; negative(positive) values represent ascent**




(descent)) at 500 hPa; (c) geopotential height (red contours, CI = 10 gpm), divergence (shading, $10^{-5}$ m s$^{-1}$) and wind

vector (vector, m s$^{-1}$) at 850 hPa.

**Figure 5: The same as Figure 5 but a composite of non-anomalous weather maps during the periods of November 9-15 and 19-21, December 6-10 and 19-25 in 2015: (a) meridional wind (black contours, CI (contour interval) = 1 m s$^{-1}$, solid (dashed) lines represent southerly (northerly) wind), divergence (shading, $10^{-5}$ m s$^{-1}$) and westerly jet (green**





line) (zonal mean wind, CI > 30 m $s^{-1}$) at 200 hPa; (b) geopotential height (black contours, CI = 50 gpm) and vertical velocity (shading, $10^{-2}$ Pa $s^{-1}$; negative(positive) values represent ascent (descent)) at 500 hPa; (c) divergence (shading, $10^{-5}$ m $s^{-1}$) and wind vector (vector, m $s^{-1}$) at 850 hPa.

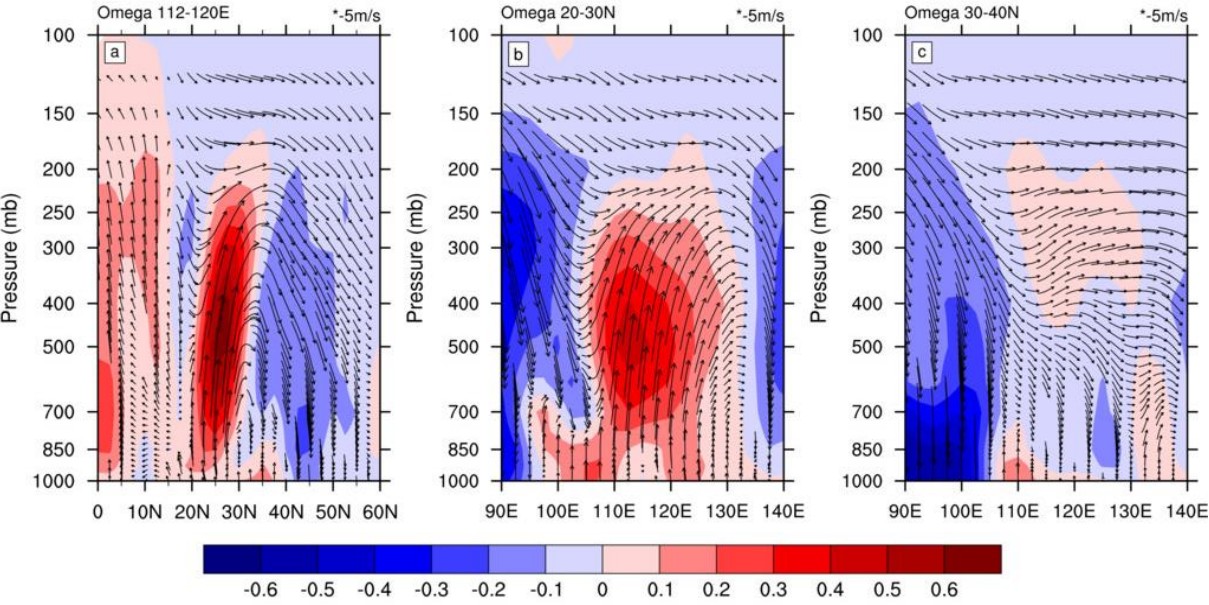


Figure 6: Latitude-height sections with average longitude in 112°E-120°E of vertical velocity (shading, unit: -5 m $s^{-1}$) and wind vector (u and ω) (a); longitude-height sections with average latitude in 20°N -30°N (b) and in 30°N-40°N (c) of vertical velocity (shading, unit: -5 m $s^{-1}$) and wind vector (u and ω) in November and December 2015.





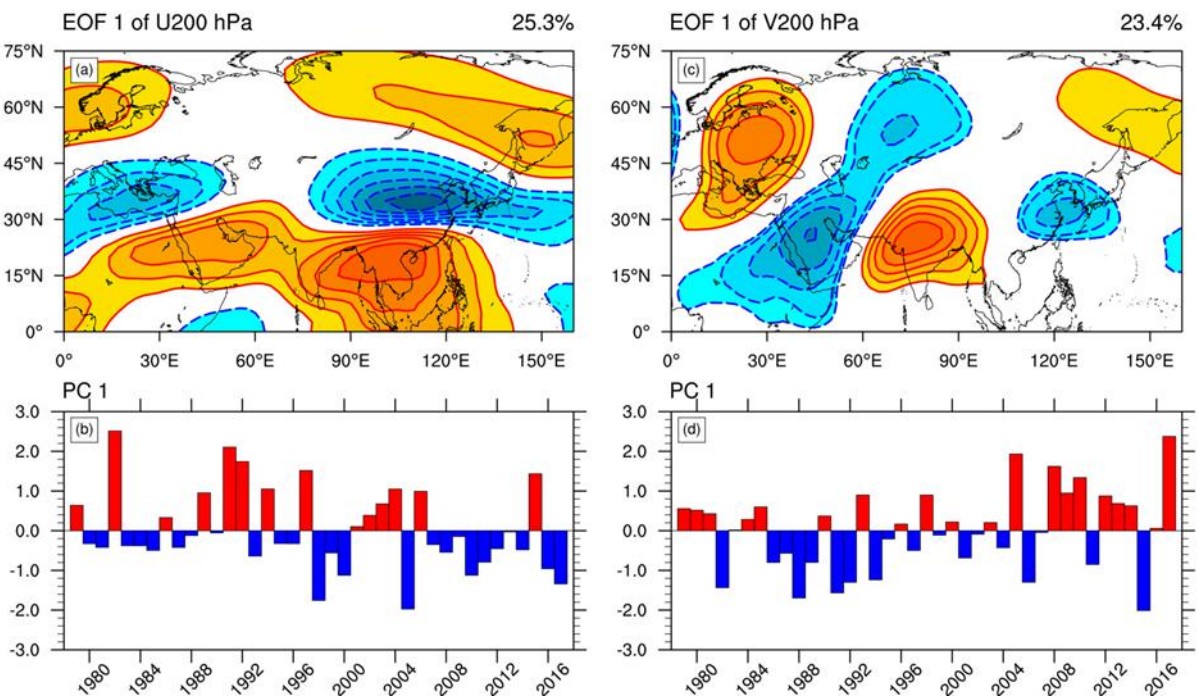

**Figure 7: The (a) spatial distribution and (b) its principal component (PC) of the leading EOF mode for 200 hPa zonal wind anomaly within the domain 0-75°N, 0-160°E in November and December 1979-2017, where the solid (dashed) lines represent westerly (easterly) winds; (c) and (d) same as (a) and (b) but for meridional wind within the domain 0-75°N, 0-160°E, where solid (dashed) lines represents southerly (northerly) winds. The percentage values in the upper-right corners of (a) and (c) show the percentage variance explained by this mode.**

Figure 8: The 200 hPa horizontal wind anomalies (vector, m s$^{-1}$) and meridional wind anomalies (contours, m s$^{-1}$) regressed onto the standardized PC1 of the leading EOF mode for 200 hPa (a) meridional and (b) zonal wind anomalies. The values of the contours and arrows are regression coefficients. The solid red (dashed blue) contours indicate the positive (negative) meridional wind anomaly. The thick black line in (a) and (b) delineates the climatological jet axis (the gradient of the zonal wind with the longitude is 0). Only anomalies statistically significant at the 0.1 level based on







**Student's t test are shown in (a) and (b).**

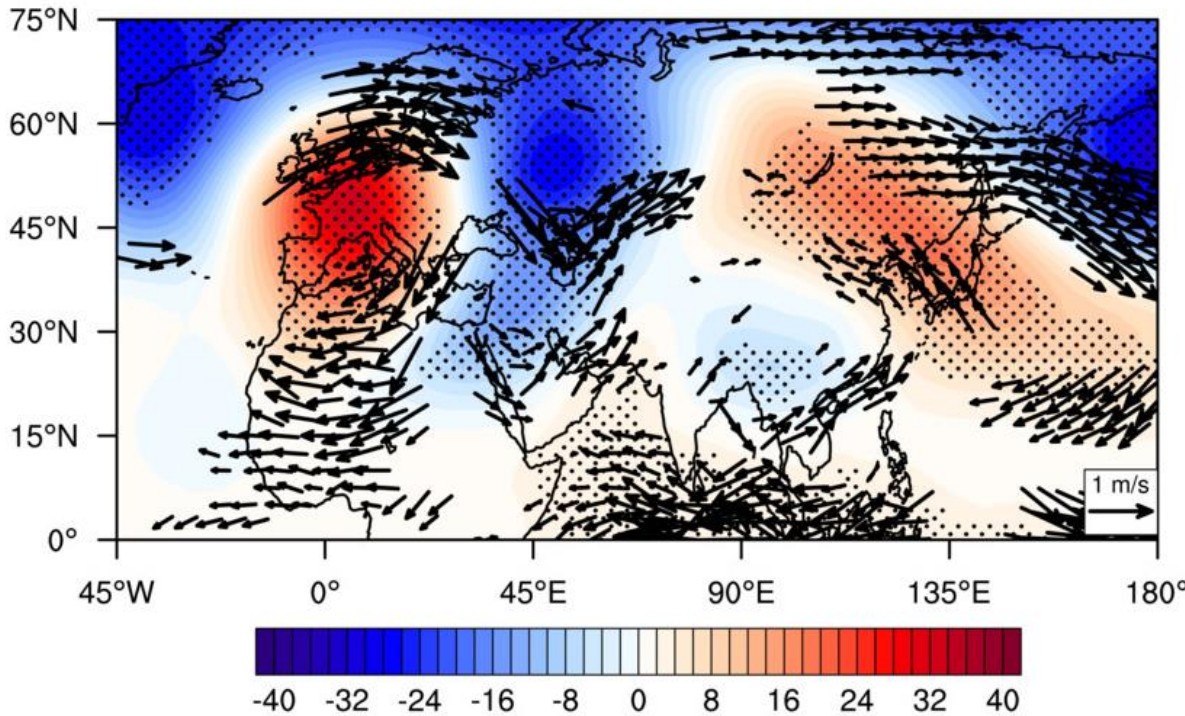

**Figure 9: Regression of 500 hPa geopotential height anomalies and 850 hPa wind vector onto the PC1 of the leading EOF mode for the 200 hPa meridional wind anomaly. The values of shading and arrows represent regression coefficients. Regions of dotted areas indicate anomalies exceeding the 0.05 confidence level. Only anomalies statistically significant at the 0.05 level based on Student's t test are given for the wind vector.**


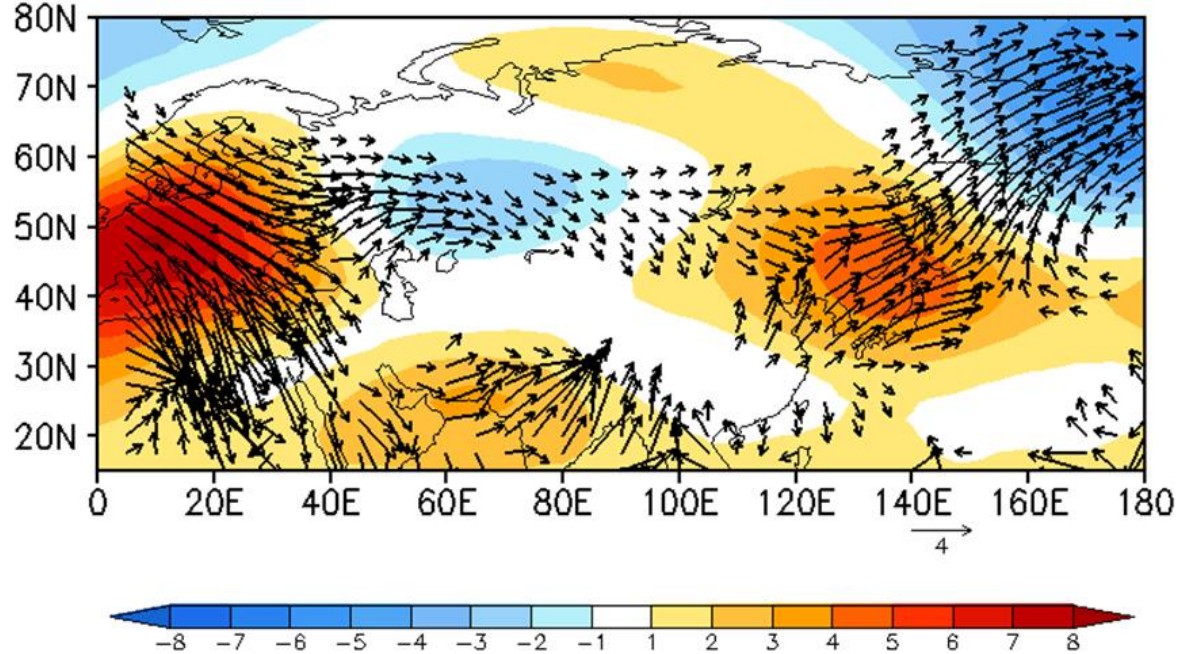

**Figure 10: Anomalous geopotential height (shading, 10 gpm) at 200 hPa in November and December 2015 and its stationary wave activity flux (vector, $m^{-2}$ $s^{-2}$).**

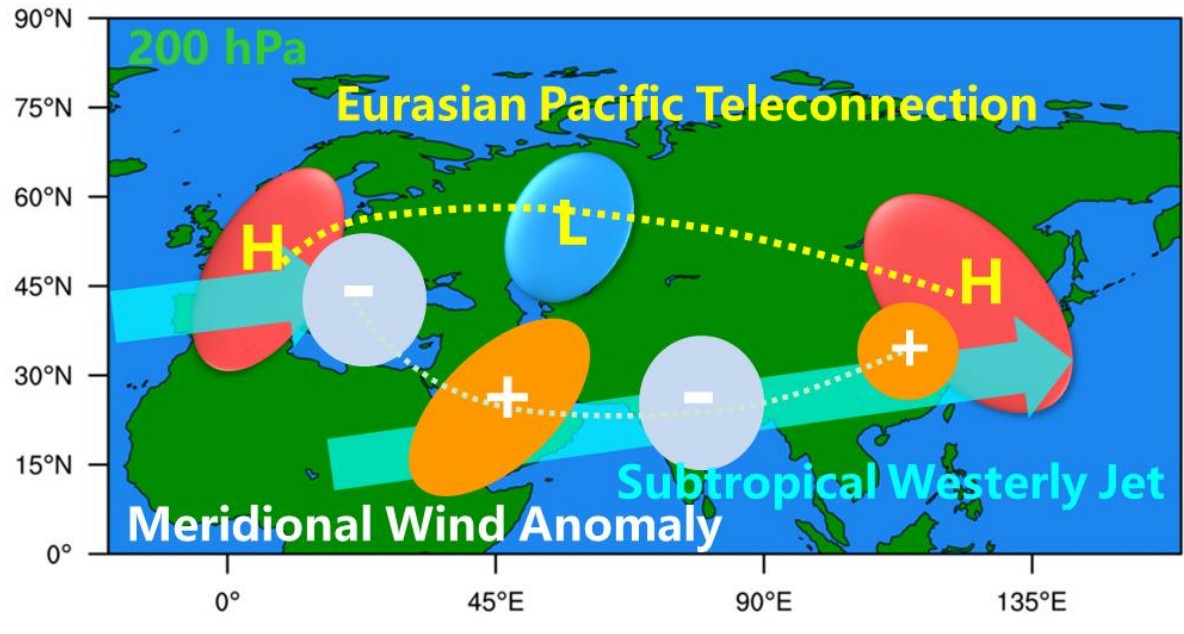


**Figure 11: A schematic diagram of the negative EU in the 200 hPa geopotential height field and meridional wind**

**anomaly at 200 hPa in November and December 2015. H (L) denotes positive (negative) geopotential height anomalies; the plus and minus represent positive (southerly) and negative (northerly) meridional wind anomalies, respectively; and the shaded belt of arrow represents the subtropical westerly jet.**

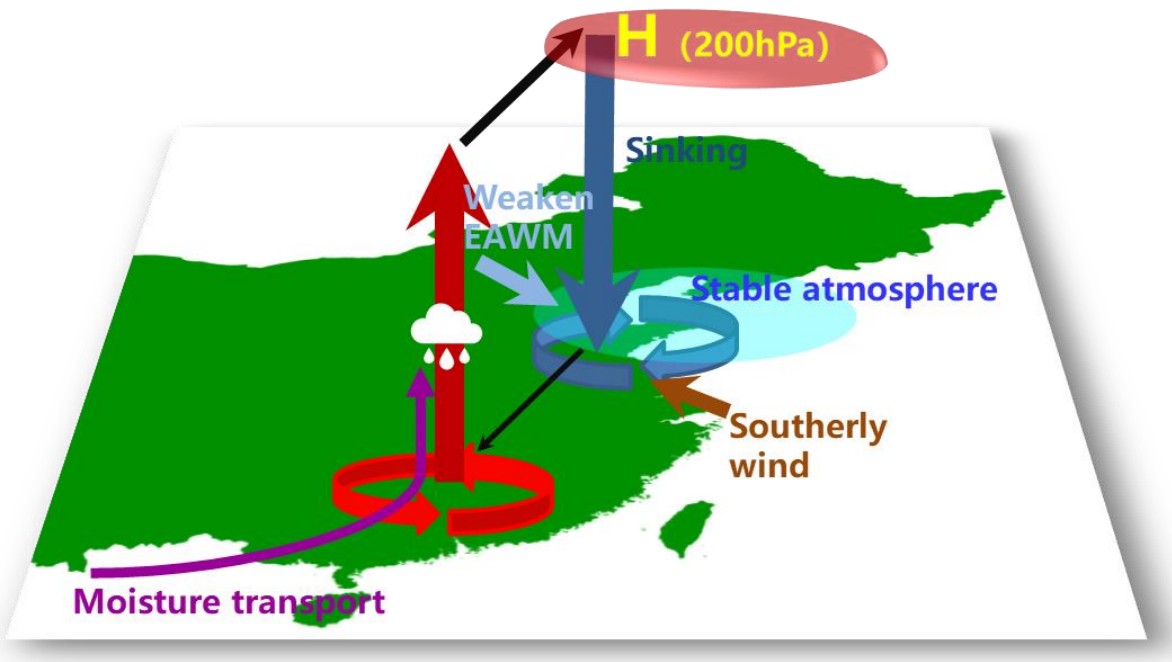


**Figure 12: Schematic illustration showing circulation system affecting the haze events in November and December 2015. The red (blue) circular of arrow represents convergence (divergence); the red (blue) translucent arrow represents ascending (descending) air; the brown translucent arrow represents the southerly wind anomaly; the purple solid arrow represents the water vapor transported by southwesterly airflow from the Bay of Bengal and South China Sea; the**

**thick (thin) black arrow represents northward (southward) movement in the atmosphere; H denotes positive geopotential height anomaly; the white weather symbol represents precipitation.**





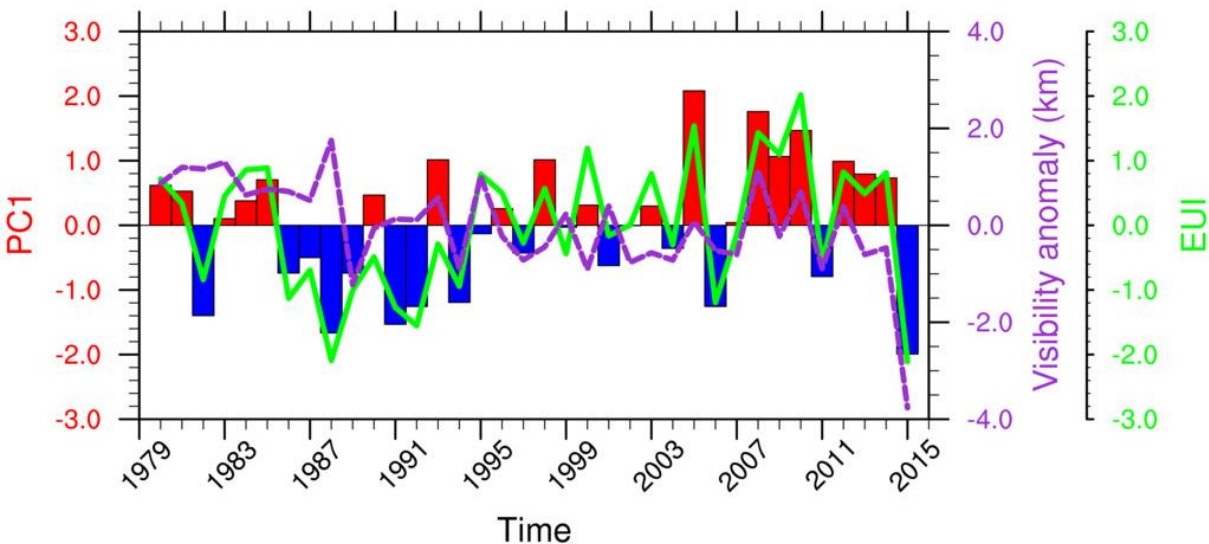

**Figure 13: The mean visibility anomaly (dashed line, purple) in November and December in the NCP, the EUI (solid line, green), and PC1 (bars, red and blue) of the leading EOF mode for the 200 hPa meridional wind anomaly from 1980 to 2015. The linear regression coefficient of the mean visibility onto PC1 is approximately 0.38, which can pass the significance test at the 95% level.**
