# Peer review of "The combined effect of two westerly jet waveguides on heavy haze in the North China Plain in November and December 2015"

_Atmospheric Chemistry and Physics, 2019_

## Referee Comment (RC1) · Anonymous Referee #2 · 14 Dec 2019

This paper is a case study of a period in November and December 2015 that was characterized by many haze days in the North China plain. The authors describe the synoptic conditions that led to an environment that was conducive to the occurrence of haze and relate this to the larger scale waveguide and quasi-stationary wave environment. Overall, it's concluded that Rossby wave trains propagating from Western Europe along two waveguides toward east Asia were responsible for setting up decent over the North China plain and a weaker than normal winter monsoon - both of which were favourable for haze. Overall, I found this to be a worthwhile study and I think it nicely describes the conditions that have led to this event. In the end, there is also some discussion of the extent to which other events have related to this kind of large

scale environment, which I think is valuable. I have only relatively minor comments to suggest before publications, although I do have quite a number of them that are mostly aimed at improving readability. I think there is some confusion throughout the text of the difference between the waveguide and the wave themselves. The authors often refer to waveguide when I think they should really be referring to the waves. I've pointed out a couple of cases below, but I suggest attention be paid to that during the revisions.

Minor comments by line number

l16: it is stated that "two Rossby waveguides within the westerly jet" are responsible for the haze. But isn't it really the Rossby waves that propagate along these waveguides that are ultimately responsible. If so, the wording could be clearer with something along the lines of "...anomalous Rossby waves that propagated along two waveguides within the westerly jet..."

l27: I would recommend simply stating "This study elucidates the formation...". Let the science speak for itself and determine whether it is of "great significance" or not.

l46: It's not clear what the anomalies here are referring to. I assume it's geopotential height, so suggest stating "In the negative EU, there is a positive anomaly in geopotential height in Europe..."

l94-95: It doesn't seem like the Cressman paper is actually cited here. Suggest "using the Cressman interpolation method (Cressman 1959)"

l103: I don't find it clear what "function of interpolation" means. Would it be clearer to state, "is the interpolated value at point i,j". Also, since the same symbol is used for latitude below, perhaps it would be better to choose a different symbol for this.

l105: I think it should be "Plumb" not "Plum"

l115: I think some more explanation of what standardization means. Does this mean they are anomalies from the mean and normalized to have standard deviation = 1?

l126: I'm confused about what 70% is referring to. 70% of the total of what? 22 days isn't 70% of the total days in November and December, so I'm not sure what this is referring to.

l133: suggest being more specific since this is only referring to the lower troposphere e.g., "so that the lower troposphere was relatively stable"

l153: Are these number of 37% and 25% referring to this particular event or to air pollution in China more generally. I think this could be stated more clearly to distinguish between the two.

l161: There are many anomalies in this figure from west to east. I suggest being more explicit about which you are referring to e.g., "there was a clear northerly anomaly in western China and southerly anomaly in eastern China in the meridional wind..."

l164: Similarly, it's not very clear what "the wave train" is referring to. Suggest pointing to Figure 4b here and describe the wave train of relevance.

l172: suggest pointing to the figures for the variables described here.

Figure 5: Why show the anomalies in figure 4 and the actual values here. It makes it difficult to compare them. Suggest that it might be more useful to show the anomalies in Figure 5 as opposed to the actual values.

l216: it doesn't make much sense that the positive phase of the EOF means that the jet is strong when above it has been stated that the EOF represents a north-south movement of the subtropical jet. Presumbly accompanying this north-south movement is an overall change in the jet strength, so I suggest stating that where the north-south movement is mentioned.

l217: Here's another place where I think there is some confusion between a Rossby wave and a Rossby waveguide. The meridional wind is associated with the Rossby wave which is propagating along the Rossby waveguide, so I suggest "waveguide" –> "wave".

l222-224: I think the wording here would be clearer as "To investigate the relationship between the position change in the jet and the Rossby waveguide, the time series of the 200hPa vector wind field and meridional wind are regressed onto the principal component time series of the first EOF of meridional wind and the first EOF of zonal wind in Figs 8 a and b, respectively"

l224: To back up this conclusion that the first EOF of meridional and zonal wind are strongly related to each other, why not just state the correlation between the principal component time series of the two EOFs?

l248: It's not clear to me that the dispersion of Rossby wave energy necessarily should result in heavy rainfall. I'm not sure to what extent these two things are always connected. Perhaps this could be clarified.

l281: A value of 0.38 quoted here sounds like it's a correlation coefficient that's being quoted as opposed to a regression coefficient. If it's a regression coefficient then there should be some units.

l288: I don't think this further shows the combined effect of the two waveguides has important influence on Haze. Doesn't it just show that the meridional wind and the geopotential height are related to each other? If so, this is to be expected, so I suggest deleting this statement.

Figure 3 caption: I think what's plotted in (a) is the difference in temperature between 1000hPa and 850hPa, but that's referred to simply as the lapse rate in the caption. I think it would be clearer to state that this is the vertical temperature difference between those levels. Suggest stating also what this is an Average over e.g., "Anomalies averaged over November and December 2015.

Figure 4b: I'm not sure if it'll make too much difference, but since the intention is to look at the wave motions, it might be clearer to plot eddy geopotential height as opposed to just geopotential height.

[Figure]

Figure 6: Best specify in the caption whether these are anomalies or not.

Figure 8 caption: line 2 suggest "(a) meridional wind and zonal wind" –> "(a) meridional wind and (b) zonal wind"

Typo's/wording:

l34: "The causes of haze in China, except for pollutant emissions,..."Aside from pollutant emissions, ..."

l70: suggest deleting "is analyzed only"

l72: "are significant" –> "were significant"

l121: "mean in November" –> "mean over November"

l128: "less 10" –> "of less than 10"

l145: "was a negative" –> "exhibited a negative"

l155: "anticyclone" –> "anticyclonic"

l168: "anomaly" –> "anomalies" (since this is referring to multiple anomalies.

l170: "circulation southern" –> "circulation in southern"

l181: "conductive" –> "conducive", but also it may be clearer to state "stabilizes the atmosphere, creating an environment that is conducive to haze"

l191: "at the 200hPa" –> "at 200hPa"

l191: "average locations" –> "averages"

l196: "intruding China" –> "intruding into China"

l197: "provides" –> "provided"

l213: I think this needs some rewording. Perhaps "To explore the influences on Rossby wave propagation along the subtropical westerly jet and associated influences on

haze..."

l228: "regressed by" –> "regressed onto"

l230-231: "the adjacent Sea of Japan areas" –> "the Sea of Japan" ?

l252: "is very weak" –> "was very weak" and "is still" –> "was still"l

l275: "of Rossby" –> "of the Rossby"
* * *

---

## Author Comment (AC1) · 17 Dec 2019

We appreciate the reviewer for carefully reviewing our manuscripts and providing the valuable suggestion to improve quality and readability for our paper. We carefully revised the manuscripts as suggested. The following are our responses to your comments point by point. The red sentences are the reviewer's comments (italics), the black sentences are the author's responses, and the blue sentences and words are the specific revisions.

**Anonymous Referee #2**

This paper is a case study of a period in November and December 2015 that was characterized by many haze days in the North China plain. The authors describe the synoptic conditions that led to an environment that was conducive to the occurrence of haze and relate this to the larger scale waveguide and quasi-stationary wave environment. Overall, it's concluded that Rossby wave trains propagating from Western Europe along two waveguides toward east Asia were responsible for setting up decent over the North China plain and a weaker than normal winter monsoon - both of which were favorable for haze. Overall, I found this to be a worthwhile study and I think it nicely describes the conditions that have led to this event. In the end, there is also some discussion of the extent to which other events have related to this kind of large-scale environment, which I think is valuable. I have only relatively minor comments to suggest before publications, although I do have quite a number of them that are mostly aimed at improving readability. I think there is some confusion throughout the text of the difference between the waveguide and the wave themselves. The authors often refer to waveguide when I think they should really be referring to the waves. I've pointed out a couple of cases below, but I suggest attention be paid to that during the revisions.

Response: We thanks for your suggestions and we have revised our manuscripts as suggested which is marked in the blue in the new manuscripts.

**Minor comments by line number**

116: it is stated that "two Rossby waveguides within the westerly jet" are responsible for the haze. But isn't it really the Rossby waves that propagate along these waveguides that are ultimately responsible. If so, the wording could be clearer with something along the lines of "...anomalous Rossby waves that propagated along two waveguides within the westerly jet..."

Response: We thank the reviewer for the suggestion. Indeed, it is the Rossby waves that propagate along two waveguides within the westerly jet. We agree with the reviewer and the sentence has been revised as suggested. In addition, we have revised some other similar statements.

115: In this paper, the combined effect of the anomalous Rossby waves within two westerly jet waveguides on...

117: ...that the anomalous Rossby waves that propagated along two waveguides within the westerly jet originating from the...

168: In summary, the Rossby waves within the East Asian upper westerly jet waveguides has important influences on the East Asian...

172-75: ...circulations in the upper troposphere on haze, especially the combined effect of the Rossby waves within two westerly jet waveguides, are rare. Given the above content, the objective of the present study is to determine whether the effects of the Rossby waves within two westerly jet waveguides on haze were significant in the NCP in November and December 2015 and, if so, to identify the principal mechanism behind the effects of the Rossby waves within two westerly jet waveguides on haze.

178: Section 4 demonstrates the influencing mechanisms of the Rossby waves within two westerly jet waveguides on haze events.

1218: 4 Evolution of the Rossby waves within two westerly jet waveguides and principal mechanism analysis

1228: To investigate the relationship between the position change in the jet and the Rossby waves, the time series of the 200hPa...

1231: Figure 8a and Figure 8b. That is, the position movement of the jet and the Rossby waves was closely related in winter, which...

1234: To reveal the influence of the Rossby waves within jet waveguide on atmospheric circulation in the middle and lower...

1246: To further examine the mechanism behind the propagation of the Rossby waves, two-dimensional wave activity flux was...

1258: flux was obviously enhanced in the Sea of Japan area, which may be due to the concurrent effect of two waves...

1261: Based on the analysis above, a diagram of Rossby waves within the waveguide of the westerly jet affecting haze events in...

1275: The Rossby waves within two westerly jet waveguides originating...

1286: The Rossby waves within two westerly jet waveguides may be the main mechanism leading to the occurrence and maintenance...

1294-300: The linear correlation coefficient of the PC1 of the leading EOF mode for the 200 hPa meridional wind anomaly and EUI is approximately 0.92, which further shows that the combined effect of the waves within two westerly jet waveguides may have an important influence on heavy haze in the NCP. In addition to 2015, the two waves also existed in 1989, 1994, 1996, 2004, 2006, and 2011. The visibility anomaly in the NCP in these years was negative. However, the waves within two westerly jet waveguides were also strong in 1982, 1986, 1988, 1991, and 1992, and the visibility anomaly in the NCP was positive (Figure 13). In addition, compared to the waves in 2015, the Rossby waves were weak or out of phase in 2000, 2002, 2003, 2007, 2013, and 2014, while the visibility anomaly in the NCP was negative.

127: I would recommend simply stating "This study elucidates the formation...". Let the science speak for itself and determine whether it is of "great significance" or not.

Response: We thank the reviewer for the suggestion. The sentence has been revised as suggested.

127: This study elucidates the formation and maintenance mechanism of large-scale haze in the NCP in late fall and boreal winter.

146: It's not clear what the anomalies here are referring to. I assume it's geopotential height, so suggest stating "In the negative EU, there is a positive anomaly in geopotential height in Europe..."

Response: We are sorry for this confusion. It is the anomalies of geopotential height. The sentence has been revised as suggested.

147-48: In the negative EU, there is a positive anomaly in geopotential height in Europe and East Asia and, a negative anomaly in Siberia at 500 hPa.

194-95: It doesn't seem like the Cressman paper is actually cited here. Suggest "using the Cressman interpolation method (Cressman 1959)"

Response: We thank the reviewer for the suggestion. The sentence has been revised as suggested.

195-96: Visibility station data were interpolated on the regular grid of  $0.5^{\circ} \times 0.5^{\circ}$  using Cressman interpolation method (Cressman, 1959).

1103: I don't find it clear what "function of interpolation" means. Would it be clearer to state, "is the interpolated value at point i,j". Also, since the same symbol is used for latitude below, perhaps it would be better to choose a different symbol for this.

Response: We agree with the reviewer. The sentence has been revised as suggested. Also, we replace  $\phi_{i,j}$  with  $S_{i,j}$  to distinguish it from latitude below.

1104: Where  $S_{i,j}$  is the interpolated value at point i, j and  $S_{obs}^k$  is station data. N is number of stations.

1105: I think it should be "Plumb" not "Plum"

Response: Yes, it is Plumb. As you suggested, the word of "Plum" been corrected into "Plumb".

1105-106: To analyze the anomalous propagation of Rossby waves, we calculated horizontal stationary wave activity flux to show the propagation of wave energy using the method of Plumb.

**1115: I think some more explanation of what standardization means. Does this mean they are anomalies from the mean and normalized to have standard deviation = 1?**

Response: We are sorry for this confusion. The standardization means they are anomalies from the mean and normalized to have standard deviation = 1 and mean value = 0. We have added formula for calculating standardized anomalies as follows:

1116:  $Norm(x) = \frac{x - mean(x)}{\sigma}$

1117: ...x is a variable.  $\sigma$  is the number of non-missing value.

1126: I'm confused about what 70% is referring to. 70% of the total of what? 22 days isn't 70% of the total days in November and December, so I'm not sure what this is referring to.

Response: We are sorry for this confusion. 22 days isn't the total days in November and December, but in November and December, respectively. So, 70% is the proportion of haze days every month. The sentence has been revised as following.

1129-130: There were 22 haze days, accounting for more than 70% of the total days, in November and December, respectively (Figure 2).

1133: suggest being more specific since this is only referring to the lower troposphere e.g., "so that the lower troposphere was relatively stable".

Response: We thank the reviewer for the suggestion. The sentence has been revised as suggested.

1136-137: The value of the static stability was small so that the lower troposphere was relatively stable (Figure 3a).

1153: Are these number of 37% and 25% referring to this particular event or to air pollution in China more generally. I think this could be stated more clearly to distinguish between the two.

Response: We thank the reviewer for the suggestion. According to Yang et al. (2016), data of daily PM2.5 concentration and daily meteorological variables in eastern China (105–122.5°E, 20–45°N) in DJF of 1985–2005 were from GEOS-4 meteorology. So, we think number of 37% and 25% referring to air pollution in China more generally. The sentence has been revised as following.

1154-157: In addition, compared with surface temperature, surface specific humidity, cloud fraction, precipitation rate, and upward flux (UP), the contributions of horizontal wind and the boundary layer height to  $PM_{2.5}$  concentration in eastern China in winter 1985-2005 were 37% and 25%, respectively (Yang et al., 2016).

1161: There are many anomalies in this figure from west to east. I suggest being more explicit about which you are referring to e.g., "there was a clear northerly anomaly in western China and southerly anomaly in eastern China in the meridional wind..."

Response: We thank the reviewer for the suggestion. The sentence has been revised as suggested.

1164-165: At 200 hPa (Figure 4a), there was a clear northerly anomaly in western China and southerly anomaly in eastern China in the meridional wind within the mid-high latitude westerly jet.

1164: Similarly, it's not very clear what "the wave train" is referring to. Suggest pointing to Figure 4b here and describe the wave train of relevance.

Response: We are sorry for this confusion. This wave train only refers to the 200hPa meridional wind anomaly (Figure 4a), not the 500hPa EU (Figure 4b). The sentence has been revised as following.

1165-169: Within the subtropical westerly jet (Figure 4a, green contour line), there existed a southerly/northerly wind wave train package which is similar to study of Ding et al. (2017) (Figure 4a). There were northerly wind anomalies in the Mediterranean region, southerly wind anomalies in the Arabian Peninsula, northerly wind anomalies in western China, and southerly wind anomalies in the Yellow and Bohai Sea of China. Hoskins et al. (1993) and Hsu et al. (1992) found that there are Rossby wave in mid latitude upper troposphere of northern hemisphere.

**1172: suggest pointing to the figures for the variables described here.**

Response: We thank the reviewer for the suggestion. We have pointed to the figures for the variables.

1175-177: The northerly and southerly wind anomalies over the Asian continent resulted in anomalous circulation southern China, with abnormal ascending movement on the eastern side of the cyclonic anomaly according to the value of divergence at 850 hPa (Figure 4c) and 200 hPa (Figure 4a) and the omega value at 500 hPa (Figure 4b).

**Figure 5: Why show the anomalies in figure 4 and the actual values here. It makes it difficult to compare them. Suggest that it might be more useful to show the anomalies in Figure 5 as opposed to the actual values.**

Response: We thank the reviewer for the suggestion. It will be more clearly to show the anomalies in Figure 5 if only to compare with Figure 4, but also to look at the background of large-scale circulation as well as the development of trough and ridge when haze occurs. We also show the anomalies (not shown in manuscripts). The anomalies are similar to those in Figure 4. There is anomalous divergence at 200hPa and anomalous convergence at 850hPa in NCP, which is not conducive to descend motion. However, the actual value shows that there is descend motion (Figure 5 in manuscripts). Besides, southerly anomaly occurred at 850hPa (Figure 4c in manuscripts). The two above indicate that the descend movement and the southerly wind anomaly caused haze, which is consistent with the results of the paper. In order to show the background of large-scale circulation when haze occurs, we use weather map of the actual value. The picture below shows the anomalies.

---

## Referee Comment (RC2) · Anonymous Referee #1 · 19 Dec 2019

Review for manuscript: acp-2019-826

The combined effect of two westerly jet waveguides on heavy haze in the North China Plain in November and December 2015

Xiadong An, Lifang Sheng, Qian Liu, Chun Li, Yang Gao, and Jianping Li

This manuscript presents an analysis of a heavy haze event in the North China Plain (NCP) in 2015, aiming to determine possible influences of the large-scale circulation on the event. It is shown that the strong haze event coincided with a wave-like pattern in the upper-tropospheric meridional wind field spanning Eurasia. It is then shown that this pattern is in fact very similar to the leading EOF of that field, and it is suggested

that this setup is vulnerable to haze events in the NCP because it is associated with (1) subsidence and increased lower-tropospheric static stability in the NCP region (forced by the polar front wave guide) and (2) additional subsidence in the NCP region resulting from ascent and diabatic heating in South China (forced by the subtropical waveguide).

This topic is of strong importance and the study helps clarify the mechanisms by which the large-scale dynamics influences local air quality in China. I do however have a number of concerns which I would like to see addressed before publication.

Main comments

- Your analysis uses mostly time-mean fields together with the Plumb stationary wave activity flux, but in the abstract you say 'The Rossby wave propagated eastward along the subtropical jet' (line 8). After reading the manuscript I am now imagining a stationary Rossby wave with an eastward group velocity (i.e. fixed ridge and trough positions, but wave activity propagating downstream), but certainly did not imagine that when I first read the abstract. Have I interpreted this correctly? Given its repeated use and importance throughout the manuscript, I wonder if there is a more precise terminology for this process? At the very least I would like to see this point clarified in the abstract and in your description of it in the manuscript.

- It is argued that it is the 'combined effect…of two Rossby waveguides' that leads to these conditions and is therefore responsible for the haze. Whilst this seems plausible, I struggled to see much direct evidence to support the second route, specifically the fact that ascent and diabatic heating in Sough China should lead to descent further north. Given that the 'combined effects' or both routes this makes it into the title, I would like to the see the argument made more precise.

Other comments

- L10. You say 'resulted in a stable atmosphere', but this is only true for the lower-troposphere. Please amend (I note Reviewer 2 also made this point later on in the

manuscript).

- L24-25. You say 'The latent heat released by rainfall acted as a heat source, inducing convection over South China'. I don't get the logic here. Do you distinguish between convective and non-convective rainfall? Where in the text do you justify the point that diabatic heating from non-convective rainfall leads to convection?

- L25-26. I think the conclusion that the ascending motion over South China helps maintain the descent over the NCP is too speculative to be stated in the abstract, given the evidence presented.

- L39. I wasn't familiar with the term 'haze-fog'. Please could you clarify its definition?

- L46. What do you mean by the 'contribution rate is 45%'? Is this the fraction of variance explained? Please clarify.

- L70. 'large' -> 'large-scale'.

- L95. I noted you have added the Cressman (1959) reference following a comment from Reviewer 2, but there is a typo in the author name.

- L104. What do you mean by 'anomalous propagation of Rossby waves'? I was not 100% sure of the methodology here. Is 'perturbation streamfunction' (L107) the same this as streamfunction anomaly (as defined in L117), or is it an anomaly from the zonal mean? Do you compute this from monthly mean fields? Please clarify the method.

- Eq 4. You have mis-placed the close bracket on the bottom line of the vector.

- L115. I noted you have added a definition of 'Norm' following a comment from Reviewer 2, but define sigma as the 'number of values'. Isn't sigma the standard deviation?

- L132. Your definition of 'static stability' is very confusing, given that the atmosphere is stable when this diagnostic is small, and unstable when it is large. Can you think of a more appropriate name for this variable, or else at least reverse its sign?

- L136. 'atmosphere' -> 'lower troposphere'

- L152. What is 'upward flux'?

- L166: You say the wind 'formed a convergence of cold air in the Middle East. . .'. Would 'led to the advection of cold air towards the Middle East. . .' be a more accurate description?

- L174. Are you referring to Fig 4c, not 4b?

- L176. You say 'There was a negative phase of the EU', but have not introduced the EU pattern anywhere.

- L180. 'atmosphere' -> lower-troposphere'

- L213. Please state the domain used for the EOF computation (it's stated in the caption, but should also appear in the main text). I'm assuming from the PC timeseries that these EOFs are based on the Nov-Dec average fields for each year (rather than monthly or daily values)? Please clarify.

- L215. You say 'The mode's variance is 25.3%'. Do you mean 'The variance explained by this mode is 25.3%'?

- L218. Same comment.

- L240. 'Plum' -> 'Plumb'

- Lines 239 to 253. You make a lot of claims regarding the interpretation of the wave activity flux plot without backing them up. E.g. 'energy dispersion over the Siberian area, leading to a negative geopotential height anomaly in this location'. Please can you provide references explaining why this is the case. Similarly why should energy 'dispersed within South China' result in a 'large amount of heavy rainfall in South China'?

- Fig 4 caption. I was confused which of the variables shown are absolute and which are anomalies (e.g. divergence in panel b). Please clarify.

- Fig 7 caption. What are the units and contour intervals in panels a and c?

- Fig 8 caption. From my understanding, the two PCs used for the regression are anti-correlated and yet the resulting patterns in panels a and b have the same signs. Have I misunderstood, or does the panel a regression use -PC1 rather than PC1?

- L480. Do you mean 'the gradient of the zonal wind with the latitude is 0'?

- Fig 9 caption. What are the units of the shaded field? Presumably this regression is again using the standardised PC1 timeseries?

- Fig 10 caption. I expected units of s-2 for the wave activity flux, given Eq 4. Please check.

- Fig 11. I find this schematic unacceptable in its current form. It shows geopotential height and meridional wind at 200 hPa, but shouldn't these be directly related via geostrophic balance? Perhaps the geopotential is shown at 500 hPa instead? In case it is correct and I have misunderstood what is shown, please clarify.

---

## Author Comment (AC2) · 9 Jan 2020

We appreciate the reviewer for carefully reviewing our manuscripts and providing the valuable suggestion to improve quality for our paper. We have carefully read your comments and revised the manuscripts as suggested. The following are our responses to your comments point by point. The red sentences are your comments (italics), and the black sentences are the author's responses, and the green sentences and words are the specific revisions.

**Anonymous Referee #1**

*The combined effect of two westerly jet waveguides on heavy haze in the North China Plain in November and December 2015*

*Xiadong An, Lifang Sheng, Qian Liu, Chun Li, Yang Gao, and Jianping Li*

*This manuscript presents an analysis of a heavy haze event in the North China Plain (NCP) in 2015, aiming to determine possible influences of the large-scale circulation on the event. It is shown that the strong haze event coincided with a wave-like pattern in the upper-tropospheric meridional wind field spanning Eurasia. It is then shown that this pattern is in fact very similar to the leading EOF of that field, and it is suggested that this setup is vulnerable to haze events in the NCP because it is associated with (1) subsidence and increased lower-tropospheric static stability in the NCP region (forced by the polar front wave guide) and (2) additional subsidence in the NCP region resulting from ascent and diabatic heating in South China (forced by the subtropical waveguide).*

*This topic is of strong importance and the study helps clarify the mechanisms by which the large-scale dynamics influences local air quality in China. I do however have a number of concerns which I would like to see addressed before publication.*

**Main comments**

*- Your analysis uses mostly time-mean fields together with the Plumb stationary wave activity flux, but in abstract you say 'The Rossby wave propagated eastward along the subtropical jet' (line 8). After reading the manuscript I am now imagining a stationary Rossby wave with an eastward group velocity (i.e. fixed ridge and trough positions, but wave activity propagating downstream), but certainly did not imagine that when I first read the abstract. Have I interpreted this correctly? Given its repeated use and importance throughout the manuscript, I wonder if there is a more precise terminology for this process? At the very least I would like to see this point clarified in the abstract and in your description of it in the manuscript.*

*- It is argued that it is the 'combined effect…of two Rossby waveguides' that leads to these conditions and is therefore responsible for the haze. Whilst this seems plausible, I struggled to see much direct evidence to support the second route, specifically the fact that ascent and diabatic heating in Sough China should lead to descent further north. Given that the 'combined effects' or both routes this makes it into the title, I would like to the see the argument made more precise.*

**Response:**

Thanks for your valuable suggestion. Yes, your interpretation is right. Our expression about the propagation of Rossby energy is similar to that of predecessors. For example, Li et al. (2015) and Ding et al. (2017) found the Rossby wave originated from northwest Europe, entered into the North Africa-Asia (NAA) westerly jet, and then propagated eastward along subtropical NAA jet. Plumb (1985) gave the three-dimensional wave activity flux of stationary Rossby wave using the conservation relation of small amplitude stationary wave propagating in zonal flow, which represents the propagation direction of wave

energy. According to Plumb's definition, the wave activity flux reflects the sustained propagation of quasi stationary wave energy to the area of East Asia and West Pacific during this period when the Eurasian Continental trough and ridge is fixed. That is to say, anomalous stationary Rossby wave with an eastward group velocity will cause circulation anomalies, and further lead to change for trough and ridge because of wave activity propagating downstream. In order to solve this confusion, we have added the following (green font) in the manuscripts to explain that the circulation anomaly caused by the eastward propagation of the anomalous Rossby wave results in haze in the NCP.

**L15-18:** ... In this paper, the combined effect of the anomalous stationary Rossby waves within two westerly jet waveguides on haze in the NCP was investigated based on visibility observational data and NCEP/NCAR reanalysis data. The results showed that the circulation anomalies in Eurasia caused by the anomalous stationary Rossby waves energy propagating along two waveguides within the westerly jet originating from the Mediterranean were responsible for haze formation in the NCP.

**L259-261:** According to the definition, the wave activity flux reflects the sustained propagation of quasi stationary wave energy to the area of East Asia and West Pacific during this period when the Eurasian Continental trough and ridge is fixed (Plumb, 1985).

**L264-265:** The propagation of the Rossby wave active flux to the downstream causes the circulation anomaly in the downstream region.

Firstly, we can find there is anomalous anticyclonic circulation form regression of the 200 hPa horizontal wind anomalies (figure 8a) and 500 hPa geopotential height anomalies (figure 9) regressed onto the standardized -PC1 of the leading EOF mode for 200 hPa meridional wind anomalies, which supported the second route leading to haze in the NCP. Besides, the direct evidence that results in further descent in North is the ascent in South China, which can be also found from figure 6. Figure 6 shows composed sections of the actual values of winds during the periods of November 9-15 and 19-21, December 6-10 and 19-25 in 2015 (We are sorry that we left out time information in figure 6 in the manuscripts) when heavy haze occurred in NCP and precipitation happened in South China. The local circulation showed in figure 6 includes the ascent in the South China and the descent in the NCP. To make sure the configuration of this ascent and descent motion, you can find that there were negative vorticity anomalies at 850hPa (figure 3d), and anomalous anticyclone and downward vertical speed at 500hPa (figure 4b) when haze occurred in North China Plain (NCP), which means that there was a descent motion in NCP. Missing information has been added in caption of figure 6.

As for diabatic heating, we try to confirm it by the change of circulation pattern with and without precipitation when haze occurred in NCP. But the result is unsatisfied. There may be the influence of other meteorological factors, because the above two situations did not occur at the same time. However, Wang et al. (2019) found that the airflow convergence primarily occurs over the southwestern portion of the western North Pacific sector, where the strongly significant and positive rainfall anomaly is triggered. The enhanced significant rainfall heating may greatly intensify the ascending motion over the western North Pacific sector and the adjacent region, resulting in subsidence over the Beijing–Tianjin–Hebei region and Northeast Asia via an anomalous local meridional cell. Simulations of an anomaly atmospheric general circulation model also supported their hypothesis (Wang et al., 2019). According to their study and figure 6, we can draw a conclusion that rainfall heating may greatly intensify the ascending motion over the South China resulting in subsidence over the NCP. We have carefully analyzed

the local circulation that ascent in South China and descend in NCP and added relevant references for diabatic heating intensifying the ascending motion in South China in our manuscripts.

**L213-215**: Figure 6 shows composed sections of the actual values during the periods of November 9-15 and 19-21, December 6-10 and 19-25 in 2015. We can find there will be a local circulation in China when there is precipitation in South China (Figure 6). This local circulation includes the ascent in the South China and the descent in NCP.

**L274-278**: Wang et al. (2019) found that the enhanced significant rainfall heating may greatly intensify the ascending motion over the western North Pacific sector and the adjacent region, resulting in subsidence over the Beijing–Tianjin–Hebei region and Northeast Asia via an anomalous local meridional cell. The latent heat released by precipitation as a heat source strengthened the abnormally ascending motion in South China, which further strengthened the anticyclone anomaly in the NCP.

[Figure]

Figure 6: Composed sections of the actual values during the periods of November 9-15 and 19-21, December 6-10 and 19-25 in 2015: latitude-height sections with average longitude in 112°E-120°E of vertical velocity (shading, unit: -5 m s$^{-1}$) and wind vector (u and ω) (a); longitude-height sections with average latitude in 20°N -30°N (b) and in 30°N-40°N (c) of vertical velocity (shading, unit: -5 m s$^{-1}$) and wind vector (u and ω).

*Other comments*

*- L10. You say 'resulted in a stable atmosphere', but this is only true for the lower-troposphere. Please amend (I note Reviewer 2 also made this point later on in the manuscript).*

**Response:** Thanks for your suggestion. Indeed, reviewer 2 also pointed this error. And we have revised it as suggested.

**L21-22:** … subsequently resulted in a stable lower-atmosphere. …

*- L24-25. You say 'The latent heat released by rainfall acted as a heat source, inducing convection over South China'. I don't get the logic here. Do you distinguish between convective and non-convective rainfall? Where in the text do you justify the point that diabatic heating from non-convective rainfall leads to convection?*

**Response:** We are sorry we didn't express our intention accurately. We just think that the rainfall may further strengthen the ascending motion due to the release of diabatic heating. Wang et al. (2019) confirmed this point, they believed that the enhanced significantly rainfall heating may greatly intensify the ascending motion over the western North Pacific. The sentences have been revised based on previous studies and your suggestions.

**L26-27:** … jet occurred in a large area of southern China. The latent heat released by rainfall acted as a heat source, intensifying the ascending motion over South China so that the descending motion over the NCP was strengthened, favoring the maintenance ...

*- L25-26. I think the conclusion that the ascending motion over South China helps maintain the descent over the NCP is too speculative to be stated in the abstract, given the evidence presented.*

**Response:** We are sorry for this confusion. We can find omega is positive anomalies (descending) in NCP and negative anomalies (ascending) in South China in November and December 2015 (figure 4b). In addition, we can also find omega is positive (descending) in NCP and negative (ascending) in South China during the periods of November 9-15 and 19-21, December 6-10 and 19-25, 2015 (there was rainfall in South China in this time). Interestingly, we find that the ascendant air in South China flows northward near 200hPa – 250hPa, and then descend in North China (figure 6a), which support the conclusion that the ascending motion over South China helps maintain the descent over the NCP. We have made the following modifications to show this evidence.

**L225-228:** … In addition, we find that the ascendant air in South China flows northward near 200hPa – 250hPa, and then descend in North China (figure 6a) when there is rainfall in South China and heavy haze in the NCP at the same period. The north-south circulation system is conducive to the maintenance of haze in the NCP, whereas large-scale precipitation occurred in southern China.

*- L39. I wasn't familiar with the term 'haze-fog'. Please could you clarify its definition?*

**Response:** We are glad to explain the term. Fog and haze are both weather phenomenon with reduced visibility. Haze is the phenomenon caused by the increase or the hygroscopic growth of aerosols at a high relative humidity (Ma et al., 2014). Fog is caused by condensation of water vapor in the air near the ground (Wu et al., 2009). Haze-fog means fog and haze appear at the same time. We give the haze definition in line 30-31 since haze is mainly discussed in our manuscripts. For a better understanding, we have added the definition of fog and haze-fog in the manuscripts.

**L31-32:** … Fog is also the phenomenon reduced visibility caused by condensation of water vapor in the air near the ground (Wu et al., 2009). When fog and haze appear at the same time, it is called haze-fog.

*L46. What do you mean by the 'contribution rate is 45%'? Is this the fraction of variance explained? Please clarify.*

**Response:** We are sorry for this confusion. It is indeed the fraction of variance. According to Li et al. (2019), based on singular

value decomposition (SVD), the EU at the 500 hPa isostatic surface is the most important pattern affecting haze-fog in northern China, and its contribution rate is 45%. The sentence has been corrected properly.

**L47-48:** Li et al. (2019) reported that the Eurasian teleconnection (EU) at 500 hPa is the most important pattern affecting haze-fog in northern China using singular value decomposition, and its fraction of variance is 45%.

*- L70. 'large' -> 'large-scale'.*

**Response:** Thanks for your suggestion. We have revised it as suggested.

**L73**: … the influence of large-scale atmospheric …

*- L95. I noted you have added the Cressman (1959) reference following a comment from Reviewer 2, but there is a typo in the author name.*

**Response:** Thanks for checking our reply in time. The author name has been revised.

**L97-98:** Visibility station data were interpolated to the regular grid of $0.5° \times 0.5°$ using Cressman interpolation method (Cressman, 1959).

*- L104. What do you mean by 'anomalous propagation of Rossby waves'? I was not 100% sure of the methodology here. Is 'perturbation streamfunction' (L107) the same this as streamfunction anomaly (as defined in L117), or is it an anomaly from the zonal mean? Do you compute this from monthly mean fields? Please clarify the method.*

**Response:** We are sorry for this confusion. We mean the propagation of anomalous Rossby waves. And 'perturbation streamfunction' (L107) is the same as streamfunction anomaly (as defined in L117). We compute this according to monthly mean fields. We have added more information for the method as follows.

**L107-108:** To analyze the propagation of anomalous Rossby waves, we calculated horizontal stationary wave activity flux based on monthly-mean data to show the propagation of wave energy using the method of Plumb (1985) (4):

**L111-112:** … Geopotential anomalies calculated according to equation 10 will be used to compute perturbation streamfunction.

*- Eq 4. You have mis-placed the close bracket on the bottom line of the vector.*

**Response:** Indeed, we have mis-placed the close bracket. The equation has been revised.

**L109:** $F = p_0 cos\phi \begin{pmatrix} \frac{1}{2a^2 cos^2\phi} \left[ (\frac{\partial \psi'}{\partial \lambda})^2 - \psi' \frac{\partial^2 \psi'}{\partial \lambda^2} \right] \\ \frac{1}{2a^2 cos^2\phi} \left( \frac{\partial \psi'}{\partial \lambda} \frac{\partial \psi'}{\partial \phi} - \psi' \frac{\partial^2 \psi'}{\partial \lambda \partial \phi} \right) \end{pmatrix}$

*- L115. I noted you have added a definition of 'Norm' following a comment from Reviewer 2, but define sigma as the 'number*

*of values'. Isn't sigma the standard deviation?*

**Response:** Indeed, we usually use sigma for standard deviation. We have replaced sigma with n to distinguish them.

**L120:** $Norm(x) = \frac{x - mean(x)}{n}$

**L121:** Here, Norm represents standardization, $x$ is a variable. n is the number of non-missing value,

*- L132. Your definition of 'static stability' is very confusing, given that the atmosphere is stable when this diagnostic is small, and unstable when it is large. Can you think of a more appropriate name for this variable, or else at least reverse its sign?*

**Response:** Thanks for your suggestion. According to suggestion from you and reviewer 2. We think it is appropriative to replace static stability with the vertical temperature difference (VTD). The sentences have been revised.

**L138-140:** To research the stability of the lower atmosphere, the vertical temperature difference (VTD) is expressed by the vertical difference in the temperature between 1000 hPa and 850 hPa (Liu et al., 2017), which is on average 4.5°C-7.5°C in the NCP. The value of the VTD was small so that the lower …

*- L136. 'atmosphere' -> 'lower troposphere'*

Response: Thanks for your suggestion. We have revised it as proposed.

**L140:** The value of the VTD was small so that the lower troposphere was relatively stable (Figure 3a).

*- L152. What is 'upward flux'?*

**Response:** According to Yang et al. (2016), the upward flux (UP, kg s$^{-1}$) in Goddard Earth-Observing System (GEOS) chemical transport model (GEOS-Chem) is associated with the convergence of aerosols. The positive regression coefficient of UP indicated the convergence and accumulation of $PM_{2.5}$, which was associated with the occurrence of haze. The sentence has been revised.

**L159:** … specific humidity, cloud fraction, precipitation rate, and upward flux (kg s$^{-1}$) associated with the convergence of aerosols, the …

*- L166: You say the wind 'formed a convergence of cold air in the Middle East…'. Would 'led to the advection of cold air towards the Middle East…' be a more accurate description?*

**Response:** We agree with you and we have revised the sentence as recommended.

**L175-176:** … strong northerly anomaly over the European continent. The strong northerly wind led to the advection of cold air towards the Middle East and the Mediterranean Sea, …

*- L174. Are you referring to Fig 4c, not 4b?*

**Response:** Yes, it is figure 4c. We have deleted figure 4b.

**L184:** … was located in the middle of the cyclonic and anticyclonic anomalies, with strong southerly winds at 850 hPa (Figure

4c),

- L176. You say 'There was a negative phase of the EU', but have not introduced the EU pattern anywhere.

**Response:** We are sorry that we did not introduce EU in our manuscripts. We've added the definition of EU.

**L185-188:** …Wallace and Gutzler (1981) pointed out the Eurasian Pattern (EU) is one of teleconnections at the 500 hPa in the northern hemisphere in the winter. The EU positive phase corresponds to the negative, positive and negative geopotential height anomalies at 500 hPa over Western Europe, Ural region and Japan, respectively. Otherwise, it is negative phase (Wallace and Gutzler, 1981).

*- L180. 'atmosphere' -> lower-troposphere'*

Response: Thanks for your suggestion. We have revised the sentence as suggested.

**L192:** Due to the anticyclonic anomaly in the lower troposphere over the Sea of Japan, …

*- L213. Please state the domain used for the EOF computation (it's stated in the caption, but should also appear in the main text). I'm assuming from the PC timeseries that these EOFs are based on the Nov-Dec average fields for each year (rather than monthly or daily values)? Please clarify.*

**Response:** Yes, the PC timeseries that these EOFs are based on the Nov-Dec average fields for each year. We have added more information in the main text.

**L230-232:** … the first leading EOF mode of the 200 hPa zonal and meridional winds within the domain 0-75°N, 0-160°E based on the average fields of November and December from 1979 to 2017 was calculated.

*- L215. You say 'The mode's variance is 25.3%'. Do you mean 'The variance explained by this mode is 25.3%'?*

**Response:** We are sorry for this confusion. Yes, the variance explained by this mode is 25.3%. For easier understanding, we have revised the sentence.

**L233:** The variance explained by this mode is 25.3%.

*- L218. Same comment.*

**Response:** we have revised the sentence.

**L237:** The variance explained by this mode is 23.4%

*- L240. 'Plum' -> 'Plumb'*

**Response:** Thanks for your suggestion. Author's name has been revised as suggested.

**L259:** … calculated and generated according to the formula for the wave action flux defined by Plumb (1985).

*- Lines 239 to 253. You make a lot of claims regarding the interpretation of the wave activity flux plot without backing them up. E.g. 'energy dispersion over the Siberian area, leading to a negative geopotential height anomaly in this location'. Please can you provide references explaining why this is the case. Similarly why should energy 'dispersed within South China' result*

*in a 'large amount of heavy rainfall in South China'?*

**Response:** We're sorry we didn't clearly state the facts. And we admit that the statement here is not very clear.

The South Branch of Rossby wave propagated eastward and break in South China, meaning that there is convergence of Rossby wave energy dispersing from the upstream, which is beneficial to the circulation for precipitation in South China. Li et al. (2015) and Ding et al. (2017) found the dispersion of Rossby wave energy near the Indochina Peninsula, which leads to heavy rainfall in Southern China in winter. Dispersion of Rossby wave energy enhanced anomalous low over the Indo-China Peninsula and high over eastern China, which made northerly wind strongest over southeastern China and induced strong divergence at the first rain day. Moreover, maintenance of the strong Rossby waveguide caused the large spatial persistent heavy rainfall (Ding et al., 2017).

The northerly Rossby wave propagates eastward along the polar front jet guide. When the wave train arrives in Siberia, it causes a negative potential height anomaly center and weakens the Siberian high because its energy is all dispersed from the upstream (Figure 10). When the wave train arrives in North China, it is a positive potential height anomaly, which weakens the East Asian Trough and is conducive to the anomalous descending in the NCP. Luo (2015) found that the north branch of the Rossby wave propagated from the North Atlantic to western Lake Baikal and was seemingly blocked by a blocking high over Lake Baikal, and the development and persistence of which was favored by the energy convergence there. There is also Rossby energy dispersing from upstream and negative anomaly of geopotential height in Siberia (Li et al., 2015; Li et al., 2019). The Rossby wave energy in the middle and upper troposphere may propagate southeastwards into the Asian anticyclonic anomaly centered over the Sea of Japan (ASJ) and its surrounding region, favoring the formation and sustainability of the ASJ and the associated air subsidence (Wang et al., 2019).

We have provided relevant references for our manuscripts.

**L264-266:** The Rossby wave in the polar front jet was stronger than that in the subtropical westerly jet. The propagation of northerly Rossby wave to the downstream will cause the anomaly of the downstream circulation (Luo, 2005; Li et al., 2015; Li et al., 2019; Wang et al., 2019).

**L273-274:** Ding et al. (2017) also found dispersion of Rossby wave energy enhanced anomalous low over the Indo-China Peninsula and high over eastern China, and maintenance of the strong Rossby waveguide caused the large spatial persistent heavy rainfall.

**L271-272:** Almost all of its energy dispersing from the upstream was converged within South China, resulting in anomalous circulation, conducive to a large amount of heavy rainfall in South China.

*- Fig 4 caption. I was confused which of the variables shown are absolute and which are anomalies (e.g. divergence in panel b). Please clarify.*

**Response:** We are sorry for this confusion. The variables shown in figure 4 are anomalies except for the zonal wind (green line). We have added information in the caption of figure 4.

Figure 4: Westerly jet and anomalous weather maps in November and December 2015: (a) meridional wind (black contours,

CI (contour interval) = 1 m s$^{-1}$, solid (dashed) lines represent southerly (northerly) wind anomaly), divergence (shading, $10^{-5}$ m s$^{-1}$) and westerly jet (green line) (zonal mean wind, CI > 30 m s$^{-1}$) at 200 hPa; (b) geopotential height (black contours, CI = 10 gpm (geopotential meters), solid (dashed) lines represent positive (negative) geopotential height anomaly) and vertical velocity (shading, $10^{-2}$ Pa s$^{-1}$); negative(positive) values represent ascent (descent) at 500 hPa; (c) geopotential height (red contours, CI = 10 gpm), divergence (shading, $10^{-5}$ m s$^{-1}$) and wind vector (vector, m s$^{-1}$) at 850 hPa. The variables shown in figure 4 are anomalies but the zonal wind is absolute (green line).

*- Fig 7 caption. What are the units and contour intervals in panels a and c?*

**Response:** We are sorry for this confusion. Contour interval is 0.01 m s$^{-1}$. We have added information in the caption of figure 7.

**L510:** … Contour interval is 0.01 m s$^{-1}$; …

*- Fig 8 caption. From my understanding, the two PCs used for the regression are anticorrelated and yet the resulting patterns in panels a and b have the same signs. Have I misunderstood, or does the panel a regression use -PC1 rather than PC1?*

**Response:** Yes, it is -PC1. Thank you for pointing out this error. We have revised it.

**L514-516:** Figure 8: The 200 hPa horizontal wind anomalies (vector, m s$^{-1}$) and meridional wind anomalies (contours, m s$^{-1}$) regressed onto the standardized -PC1 of the leading EOF mode for 200 hPa meridional wind anomalies (a) and the standardized PC1 of the leading EOF mode for 200 hPa zonal wind anomalies (b). …

*- L480. Do you mean 'the gradient of the zonal wind with the latitude is 0'?*

**Response:** The position of 200hPa jet stream is the place with the maximum value of zonal wind speed in the upper troposphere. The wind speed decreases with the increase of latitude in the north of the jet stream. And the wind speed decreases with the decrease of latitude in the south of the jet stream. Thus, the position where the gradient of zonal wind speed is 0, called the jet axis.

*- Fig 9 caption. What are the units of the shaded field? Presumably this regression is again using the standardised PC1 timeseries?*

**Response:** We're sorry that we didn't provide the units of the shaded field. The units of the shaded field is geopotential meters (gpm). And we use the negative standardized PC1 timeseries to regress geopotential height anomalies. We have added units of the shaded field and revised information as suggested.

**L522-524:** Figure 9: Regression of 500 hPa geopotential height anomalies and 850 hPa wind vector onto the negative standardized -PC1 of the leading EOF mode for the 200 hPa meridional wind anomaly. The values of shading (units, gpm) and arrows (units, m/s) ...

*- Fig 10 caption. I expected units of s-2 for the wave activity flux, given Eq 4. Please check.*

**Response:** Thanks for your suggestion. We have carefully check it. Because it is vector flux, units of the wave activity flux is m$^{-2}$ s$^{-2}$ depending on Eq 4, which have also added in Eq 4.

**L110:** Here, $F$ is the horizontal stationary wave activity flux (m$^{-2}$ s$^{-2}$), …

*- Fig 11. I find this schematic unacceptable in its current form. It shows geopotential height and meridional wind at 200 hPa, but shouldn't these be directly related via geostrophic balance? Perhaps the geopotential is shown at 500 hPa instead? In case it is correct and I have misunderstood what is shown, please clarify.*

**Response:** Thanks for your suggestion. As a matter of fact, geopotential should be at 500 hPa. However, it should be noted that the potential height anomaly at 200 hPa is similar to this as a result of the barotropic structure (not shown). We have revised the picture as suggested.

[Figure]

Figure 11: A schematic diagram of the negative EU in the 500 hPa geopotential height field and meridional wind anomaly at 200 hPa in November and December 2015. H (L) denotes positive (negative) geopotential height anomalies; the plus and minus represent positive (southerly) and negative (northerly) meridional wind anomalies, respectively; and the shaded belt of arrow represents the subtropical westerly jet.

Wu, G. X., and Liu, H.: The effect of spatially nonuniform heating on the formation and variation of subtropical high part I. Scale analysis, Acta. Meteorological Sinica., 57, 3, 1999.

[revised manuscript text omitted]

---

## Author Comment (AC3) · 9 Jan 2020

We appreciate the reviewer for carefully reviewing our manuscripts and providing the valuable suggestion to improve quality and readability for our paper. We carefully revised the manuscripts as suggested. The following are our responses to your comments point by point. The red sentences are the reviewer's comments (italics), the black sentences are the author's responses, and the blue sentences and words are the specific revisions.

**Anonymous Referee #2**

*This paper is a case study of a period in November and December 2015 that was characterized by many haze days in the North China plain. The authors describe the synoptic conditions that led to an environment that was conducive to the occurrence of haze and relate this to the larger scale waveguide and quasi-stationary wave environment. Overall, it's concluded that Rossby wave trains propagating from Western Europe along two waveguides toward east Asia were responsible for setting up decent over the North China plain and a weaker than normal winter monsoon - both of which were favorable for haze. Overall, I found this to be a worthwhile study and I think it nicely describes the conditions that have led to this event. In the end, there is also some discussion of the extent to which other events have related to this kind of large-scale environment, which I think is valuable. I have only relatively minor comments to suggest before publications, although I do have quite a number of them that are mostly aimed at improving readability. I think there is some confusion throughout the text of the difference between the waveguide and the wave themselves. The authors often refer to waveguide when I think they should really be referring to the waves. I've pointed out a couple of cases below, but I suggest attention be paid to that during the revisions.*

Response: We thanks for your suggestions and we have revised our manuscripts as suggested which is marked in the blue in the new manuscripts.

*Minor comments by line number*

*l16: it is stated that "two Rossby waveguides within the westerly jet" are responsible for the haze. But isn't it really the Rossby waves that propagate along these waveguides that are ultimately responsible. If so, the wording could be clearer with something along the lines of "...anomalous Rossby waves that propagated along two waveguides within the westerly jet..."*

Response: We thank the reviewer for the suggestion. Indeed, it is the Rossby waves that propagate along two waveguides within the westerly jet. We agree with the reviewer and the sentence has been revised as suggested. In addition, we have revised some other similar statements.

l15: In this paper, the combined effect of the anomalous Rossby waves within two westerly jet waveguides on…

l17: …that the anomalous Rossby waves that propagated along two waveguides within the westerly jet originating from the...

l68: In summary, the Rossby waves within the East Asian upper westerly jet waveguides has important influences on the East Asian...

l72-75: …circulations in the upper troposphere on haze, especially the combined effect of the Rossby waves within two westerly jet waveguides, are rare. Given the above content, the objective of the present study is to determine whether the effects of the Rossby waves within two westerly jet waveguides on haze were significant in the NCP in November and December 2015 and, if so, to identify the principal mechanism behind the effects of the Rossby waves within two westerly jet waveguides on haze.

l78: Section 4 demonstrates the influencing mechanisms of the Rossby waves within two westerly jet waveguides on haze events.

l218: 4 Evolution of the Rossby waves within two westerly jet waveguides and principal mechanism analysis

l228: To investigate the relationship between the position change in the jet and the Rossby waves, the time series of the 200hPa…

l231: Figure 8a and Figure 8b. That is, the position movement of the jet and the Rossby waves was closely related in winter, which…

l234: To reveal the influence of the Rossby waves within jet waveguide on atmospheric circulation in the middle and lower…

l246: To further examine the mechanism behind the propagation of the Rossby waves, two-dimensional wave activity flux was…

l258: flux was obviously enhanced in the Sea of Japan area, which may be due to the concurrent effect of two waves…

l261: Based on the analysis above, a diagram of Rossby waves within the waveguide of the westerly jet affecting haze events in…

l275: The Rossby waves within two westerly jet waveguides originating…

l286: The Rossby waves within two westerly jet waveguides may be the main mechanism leading to the occurrence and maintenance…

l294-300: The linear correlation coefficient of the PC1 of the leading EOF mode for the 200 hPa meridional wind anomaly and EUI is approximately 0.92, which further shows that the combined effect of the waves within two westerly jet waveguides may have an important influence on heavy haze in the NCP. In addition to 2015, the two waves also existed in 1989, 1994, 1996, 2004, 2006, and 2011. The visibility anomaly in the NCP in these years was negative. However, the waves within two westerly jet waveguides were also strong in 1982, 1986, 1988, 1991, and 1992, and the visibility anomaly in the NCP was positive (Figure 13). In addition, compared to the waves in 2015, the Rossby waves were weak or out of phase in 2000, 2002, 2003, 2007, 2013, and 2014, while the visibility anomaly in the NCP was negative.

*l27: I would recommend simply stating "This study elucidates the formation...". Let the science speak for itself and determine whether it is of "great significance" or not.*

Response: We thank the reviewer for the suggestion. The sentence has been revised as suggested.

l27: This study elucidates the formation and maintenance mechanism of large-scale haze in the NCP in late fall and boreal winter.

*l46: It's not clear what the anomalies here are referring to. I assume it's geopotential height, so suggest stating "In the negative EU, there is a positive anomaly in geopotential height in Europe..."*

Response: We are sorry for this confusion. It is the anomalies of geopotential height. The sentence has been revised as suggested.

l47-48: In the negative EU, there is a positive anomaly in geopotential height in Europe and East Asia and, a negative anomaly in Siberia at 500 hPa.

*l94-95: It doesn't seem like the Cressman paper is actually cited here. Suggest "using the Cressman interpolation method (Cressman 1959)"*

Response: We thank the reviewer for the suggestion. The sentence has been revised as suggested.

l95-96: Visibility station data were interpolated on the regular grid of 0.5°× 0.5° using Cressman interpolation method (Cressman, 1959).

*l103: I don't find it clear what "function of interpolation" means. Would it be clearer to state, "is the interpolated value at point i,j". Also, since the same symbol is used for latitude below, perhaps it would be better to choose a different symbol for this.*

Response: We agree with the reviewer. The sentence has been revised as suggested. Also, we replace $\emptyset_{i,j}$ with $S_{i,j}$ to distinguish it from latitude below.

l104: Where $S_{i,j}$ is the interpolated value at point i, j and $S_{obs}^{k}$ is station data. N is number of stations.

*l105: I think it should be "Plumb" not "Plum"*

Response: Yes, it is Plumb. As you suggested, the word of "Plum" been corrected into "Plumb".

l105-106: To analyze the anomalous propagation of Rossby waves, we calculated horizontal stationary wave activity flux to show the propagation of wave energy using the method of Plumb.

*l115: I think some more explanation of what standardization means. Does this mean they are anomalies from the mean and normalized to have standard deviation = 1?*

Response: We are sorry for this confusion. The standardization means they are anomalies from the mean and normalized to have standard deviation = 1 and mean value = 0. We have added formula for calculating standardized anomalies as follows:

l116: $Norm(x) = \frac{x - mean(x)}{\sigma}$

l117: …$x$ is a variable. $\sigma$ is the number of non-missing value.

*l126: I'm confused about what 70% is referring to. 70% of the total of what? 22 days isn't 70% of the total days in November and December, so I'm not sure what this is referring to.*

Response: We are sorry for this confusion. 22 days isn't the total days in November and December, but in November and December, respectively. So, 70% is the proportion of haze days every month. The sentence has been revised as following.

l129-130: There were 22 haze days, accounting for more than 70% of the total days, in November and December, respectively (Figure 2).

*l133: suggest being more specific since this is only referring to the lower troposphere e.g., "so that the lower troposphere was relatively stable".*

Response: We thank the reviewer for the suggestion. The sentence has been revised as suggested.

l136-137: The value of the static stability was small so that the lower troposphere was relatively stable (Figure 3a).

*l153: Are these number of 37% and 25% referring to this particular event or to air pollution in China more generally. I think this could be stated more clearly to distinguish between the two.*

Response: We thank the reviewer for the suggestion. According to Yang et al. (2016), data of daily PM2.5 concentration and daily meteorological variables in eastern China (105–122.5°E, 20–45°N) in DJF of 1985–2005 were from GEOS-4 meteorology. So, we think number of 37% and 25% referring to air pollution in China more generally. The sentence has been revised as following.

l154-157: In addition, compared with surface temperature, surface specific humidity, cloud fraction, precipitation rate, and upward flux (UP), the contributions of horizontal wind and the boundary layer height to $PM_{2.5}$ concentration in eastern China in winter 1985-2005 were 37% and 25%, respectively (Yang et al., 2016).

*l161: There are many anomalies in this figure from west to east. I suggest being more explicit about which you are referring to e.g., "there was a clear northerly anomaly in western China and southerly anomaly in eastern China in the meridional wind..."*

Response: We thank the reviewer for the suggestion. The sentence has been revised as suggested.

l164-165: At 200 hPa (Figure 4a), there was a clear northerly anomaly in western China and southerly anomaly in eastern China in the meridional wind within the mid-high latitude westerly jet.

*l164: Similarly, it's not very clear what "the wave train" is referring to. Suggest pointing to Figure 4b here and describe the wave train of relevance.*

Response: We are sorry for this confusion. This wave train only refers to the 200hPa meridional wind anomaly (Figure 4a), not the 500hPa EU (Figure 4b). The sentence has been revised as following.

l165-169: Within the subtropical westerly jet (Figure 4a, green contour line), there existed a southerly/northerly wind wave train package which is similar to study of Ding et al. (2017) (Figure 4a). There were northerly wind anomalies in the Mediterranean region, southerly wind anomalies in the Arabian Peninsula, northerly wind anomalies in western China, and southerly wind anomalies in the Yellow and Bohai Sea of China. Hoskins et al. (1993) and Hsu et al. (1992) found that there are Rossby wave in mid latitude upper troposphere of northern hemisphere.

*l172: suggest pointing to the figures for the variables described here.*

Response: We thank the reviewer for the suggestion. We have pointed to the figures for the variables.

l175-177: The northerly and southerly wind anomalies over the Asian continent resulted in anomalous circulation southern China, with abnormal ascending movement on the eastern side of the cyclonic anomaly according to the value of divergence at 850 hPa (Figure 4c) and 200 hPa (Figure 4a) and the omega value at 500 hPa (Figure 4b).

*Figure 5: Why show the anomalies in figure 4 and the actual values here. It makes it difficult to compare them. Suggest that it might be more useful to show the anomalies in Figure 5 as opposed to the actual values.*

Response: We thank the reviewer for the suggestion. It will be more clearly to show the anomalies in Figure 5 if only to compare with figure 4. Our aim is not only to compare with Figure 4, but also to look at the background of large-scale circulation as well as the development of trough and ridge when haze occurs. We also show the anomalies (not shown in manuscripts). The anomalies are similar to those in Figure 4. There is anomalous divergence at 200hPa and anomalous convergence at 850hPa in NCP, which is not conducive to descend motion. However, the actual value shows that there is descend motion (Figure 5 in manuscripts). Besides, southerly anomaly occurred at 850hPa (Figure 4c in manuscripts). The two above indicate that the descend movement and the southerly wind anomaly caused haze, which is consistent with the results of the paper. In order to show the background of large-scale circulation when haze occurs, we use weather map of the actual value. The picture below shows the anomalies.

[Figure]

**Figure**: The same as Figure 4 but a composite of anomalous weather maps during the periods of November 9-15 and 19-21, December 6-10 and 19-25 in 2015: (a) meridional wind (black contours, CI (contour interval) = 1 m s$^{-1}$, solid (dashed) lines represent southerly (northerly) wind), divergence (shading, $10^{-5}$ m s$^{-1}$) and westerly jet (green line) (zonal mean wind, CI > 30 m s$^{-1}$) at 200 hPa; (b) geopotential height (black contours, CI = 10 gpm) and vertical velocity (shading, $10^{-2}$ Pa s$^{-1}$); negative(positive) values represent ascent (descent) at 500hPa; (c) divergence (shading, $10^{-5}$ m s$^{-1}$) and wind vector (vector, m s$^{-1}$) at 850hPa.

*l216: it doesn't make much sense that the positive phase of the EOF means that the jet is strong when above it has been stated that the EOF represents a north-south movement of the subtropical jet. Presumably accompanying this north-south movement is an overall change in the jet strength, so I suggest stating that where the north-south movement is mentioned.*

Response: We thank the reviewer for the suggestion. And we think suggestion is very well. The sentence has been revised as suggested.

l222-223: From the first modal time series, it can be seen that the exponent is positive in 2015, meaning that the location of the subtropical jet is south.

*l217: Here's another place where I think there is some confusion between a Rossby wave and a Rossby waveguide. The meridional wind is associated with the Rossby wave which is propagating along the Rossby waveguide, so I suggest "waveguide" –>"wave".*

Response: We are sorry for this confusion and we thank the reviewer for the suggestion. The word has been revised as suggested.

l217: The first mode of the meridional wind is a Rossby wave manifested by the north-south anomaly of the 200 hPa meridional wind, following Li et al. (2017).

*l222-224: I think the wording here would be clearer as "To investigate the relationship between the position change in the jet and the Rossby waveguide, the time series of the 200hPa vector wind field and meridional wind are regressed onto the principal component time series of the first EOF of meridional wind and the first EOF of zonal wind in Figs 8 a and b, respectively"*

Response: We thank the reviewer for the suggestion. The sentence has been revised as suggested.

l222-224: To investigate the relationship between the position change in the jet and the Rossby waveguide, the time series of the 200hPa vector wind field and meridional wind are regressed onto the principal component time series of the first EOF of meridional wind and the first EOF of zonal wind in Figs 8 a and b, respectively.

*l224: To back up this conclusion that the first EOF of meridional and zonal wind are strongly related to each other, why not just state the correlation between the principal component time series of the two EOFs?*

Response: We thank the reviewer for the suggestion. Yes, it will be good to find correlation between the principal component time series of EOF. In our article, we not only want to show the correlation between the principal component time series of the two EOFs, but also want to know whether they will really cause the anomalous circulation at 200hPa corresponding to haze days mentioned above. So, we use the method of regression. we can clearly see the anomalous wind field at 200hPa by using regression analysis.

*l248: It's not clear to me that the dispersion of Rossby wave energy necessarily should result in heavy rainfall. I'm not sure to what extent these two things are always connected. Perhaps this could be clarified.*

Response: First of all, we're sorry for this confusion. We admit that the statement here is not right. The South Branch of Rossby wave propagated eastward and break in South China, meaning that there is convergence of Rossby wave energy here, which may be beneficial to the circulation for precipitation in South China. For example, Li et al. (2015) and Ding et al. (2017) found the convergence of Rossby wave energy lead to heavy rainfall in Southern China in winter. We have revised the original expression.

l254-256: Almost all of its energy was converged within South China, resulting in anomalous circulation, conducive to a large

amount of heavy rainfall in South China.

*l281: A value of 0.38 quoted here sounds like it's a correlation coefficient that's being quoted as opposed to a regression coefficient. If it's a regression coefficient then there should be some units.*

Response: Yes, it's a regression coefficient with units km. We have added units of regression coefficient in manuscript.

l289: …approximately 0.38 km from 1980 to 2015 (Figure 13).

*l288: I don't think this further shows the combined effect of the two waveguides has important influence on Haze. Doesn't it just show that the meridional wind and the geopotential height are related to each other? If so, this is to be expected, so I suggest deleting this statement.*

Response: We are sorry for this confusion. The 200hPa meridional wind anomaly according to EOF is the South Branch wave train propagating along the subtropical westerly jet expressed with PC1, while the 500hPa geopotential height anomaly is the North Branch wave train propagating along the polar front jet expressed with EUI. The correlation coefficient between PC1 and EUI is approximately 0.92, meaning that the two waveguides have important influence on haze.

*Figure 3 caption: I think what's plotted in (a) is the difference in temperature between 1000hPa and 850hPa, but that's referred to simply as the lapse rate in the caption. I think it would be clearer to state that this is the vertical temperature difference between those levels. Suggest stating also what this is an Average over e.g., "Anomalies averaged over November and December 2015.*

Response: We thank the reviewer for the suggestion. Yes, what's plotted in (a) is the vertical temperature difference between 1000hPa and 850hPa. Besides, in order to reflect the real background field, figure 3a, 3b,3c,3d is the actual values rather than the anomalies. But Figure 3e is anomalies. The caption has been revised as suggested.

Figure 3: Spatial distribution of average over November and December 2015: the vertical temperature difference between 1000hPa and 850hPa (unit: °C; Figure 3a); relative humidity at 925 hPa (unit: %; Figure 3b); wind vector (unit: m $s^{-1}$) at 10 m (Figure 3c) and 850hPa (Figure 3d) in NCP, with shading indicating the wind speed in the appropriate level. Vertical distribution of anomalies averaged over November and December 2015: air temperature (AT) (blue dashed line), horizontal wind speed (WS) (red solid line) and relative humidity (RH) (red dash-dotted line) in NCP (Figure 3e).

*Figure 4b: I'm not sure if it'll make too much difference, but since the intention is to look at the wave motions, it might be clearer to plot eddy geopotential height as opposed to just geopotential height.*

Response: We are sorry for this confusion. And we admit it might be clearer to look at the wave motions by using eddy geopotential height as opposed to just geopotential height. The aim of using geopotential height instead of eddy geopotential height here is not only to look at the wave motions, but also to look at the circulation situation at 500hPa conducive to the formation of haze. For example, according to the results in our paper, the anticyclonic anomaly at 500hPa in Sea of Japan and the cyclonic anomaly in South China are conducive to the formation of haze in the North China Plain. Besides, geopotential height anomalies at 500hPa was used to study EU pattern in many previous researches. We think that geopotential height anomalies may show more information that we need.

*Figure 6: Best specify in the caption whether these are anomalies or not.*

Response: We are sorry for this confusion. These are the actual values, not anomalies in Figure 6. The detail information has been added in Figure 6.

Figure 6: Composed sections of the actual values during the periods of November 9-15 and 19-21, December 6-10 and 19-25 in 2015: latitude-height sections with average longitude in 112°E-120°E of vertical velocity (shading, unit: -5 m $s^{-1}$) and wind vector (u and ω) (a); longitude-height sections with average latitude in 20°N -30°N (b) and in 30°N-40°N (c) of vertical velocity (shading, unit: -5 m $s^{-1}$) and wind vector (u and ω).

*Figure 8 caption: line 2 suggest "(a) meridional wind and zonal wind" –> "(a) meridional wind and (b) zonal wind"*

Response: We thank the reviewer for the suggestion. We have noticed this error. We are sorry that it's not updated in the latest manuscript. The sentence has been revised as suggested.

Figure 8: The 200 hPa horizontal wind anomalies (vector, m $\mathbf{s^{-1}}$) and meridional wind anomalies (contours, m $\mathbf{s^{-1}}$) regressed onto the standardized PC1 of the leading EOF mode for 200 hPa (a) meridional and (b) zonal wind anomalies. The values of the contours and arrows are regression coefficients. The solid red (dashed blue) contours indicate the positive (negative) meridional wind anomaly. The thick black line in (a) and (b) delineates the climatological jet axis (the gradient of the zonal wind with the longitude is 0). Only anomalies statistically significant at the 0.1 level based on Student's t test are shown in (a) and (b).

Typo's/wording:

*l34: "The causes of haze in China, except for pollutant emissions, ..."Aside from pollutant emissions, ..."*

Response: The sentence has been revised as suggested.

l34: Aside from pollutant emissions…

*l70: suggest deleting "is analyzed only"*

Response: We thank the reviewer for the suggestion. We have deleted it as following.

l69-73: Previous research on the meteorological parameters influencing haze in the NCP has focused mainly on local meteorological conditions in the middle and lower troposphere, or only the correlation between haze in China and North Atlantic SST or other patterns, while studies about the specific mechanisms behind the influence of large atmospheric circulations in the upper troposphere on haze, especially the combined effect of two westerly jet waveguides, are rare.

*l72: "are significant" –> "were significant"*

Response: The sentence has been revised as suggested.

l74: …on haze were significant in the NCP …

*l121: "mean in November" –> "mean over November"*

Response: The sentence has been revised as suggested.

l123-124: Here, $hgt\_a_{Nov2015}$, $hgt_{Nov2015}$ and $hgt\_m_{Nov\ during\ 1981\ to\ 2010}$ represent the geopotential height anomaly in November 2015, the geopotential height in November 2015 and geopotential height mean over November 1981 to 2010.

*l128: "less 10" –> "of less than 10"*

Response: The sentence has been revised as suggested.

l131-132: A regional mean visibility of less than 10 km appeared on 4-16, 18-22 and 27-30, with minimal daily values of 5.16 km, 5.86 km and 4.14 km, respectively.

*l145: "was a negative" –> "exhibited a negative"*

Response: The sentence has been revised as suggested.

l148-149: The horizontal wind speed exhibited a negative anomaly from the lower troposphere to the upper troposphere (Figure 3e).

*l155: "anticyclone" –> "anticyclonic"*

Response: The word has been revised as suggested.

l158-159: From the 850 hPa wind vector, it can be found that there was an anticyclonic circulation in southern China.

*l168: "anomaly" –> "anomalies" (since this is referring to multiple anomalies.*

Response: Yes, we agree with the reviewer. The word has been revised as suggested.

l173-175: The centers of the southerly wind anomalies were located over the Arabian Peninsula and East Asia, and the centers of the northerly wind anomaly were located over the European continent and Central Asia.

*l170: "circulation southern" –> "circulation in southern"*

Response: The sentence has been revised as suggested.

l175-176: The northerly and southerly wind anomalies over the Asian continent resulted in anomalous circulation in southern China…

*l181: "conductive" –> "conducive", but also it may be clearer to state "stabilizes the atmosphere, creating an environment that is conducive to haze"*

Response: We thank the reviewer for the suggestion. The sentence has been revised as suggested.

Due to the anticyclonic anomaly in the troposphere over the Sea of Japan, the descending movement over the NCP stabilizes the atmosphere, creating an environment that is conducive to haze.

*l191: "at the 200hPa" –> "at 200hPa"*

Response: The sentence has been revised as suggested.

l197: …anomalous meridional wind at 200 hPa…

*l191: "average locations" –> "averages"*

Response: The sentence has been revised as suggested.

l197: …monthly averages (Figure 4a, 4c and Figure 5a, 5c)…

*l196: "intruding China" –> "intruding into China"*

Response: We admit it's missing "into". The sentence has been revised as suggested.

l202: This circulation situation reduced the amount of cold air intruding into China and the winter monsoon also weakened.

*l197: "provides" –> "provided"*

Response: The word has been revised as suggested.

l203: The Southern Branch trough provided rising…

*l213: I think this needs some rewording. Perhaps "To explore the influences on Rossby wave propagation along the subtropical westerly jet and associated influences on haze..."*

Response: We thank the reviewer for the suggestion. The sentence has been revised as suggested.

l219: To explore the influences on Rossby wave propagation along the subtropical westerly jet and associated influences on haze…

*l228: "regressed by" –> "regressed onto"*

Response: The sentence has been revised as suggested.

l235: …were regressed onto the time series…

*l230-231: "the adjacent Sea of Japan areas" –> "the Sea of Japan"?*

Response: Yes, it is the Sea of Japan. The sentence has been revised as suggested.

l237-238: The two positive centers of the regression coefficients of the geopotential height anomaly are located over the

Mediterranean Sea and the Sea of Japan.

*l252: "is very weak" –> "was very weak" and "is still" –> "was still"*

Response: We thank the reviewer for the suggestion. The sentence has been revised as suggested.

l258-260: Thus, the energy from the southern branch of the Rossby wave was very weak over the Sea of Japan, but the anticyclone anomaly over the Sea of Japan was still very strong.

Response: We are sorry that it's missing "the". The sentence has been revised as suggested.

[revised manuscript text omitted]

---

## Author Response (AR2)

Dear editor,

We appreciate you for carefully reviewing our manuscripts and providing the suggestion to improve the quality of it. We have carefully read your comments and revised the manuscripts as suggested. The following are our responses to your confusions point by point. We marked all relevant changes in the manuscript in red.

*L22: Is this the wave propagating along the polar front waveguide or the subtropical wave guide or both? Please clarify*

**Response:** We are sorry we didn't express our intention accurately. This wave propagated along the polar front waveguide. The sentences have been revised.

*L84-86: how many stations are there? Where are they located? This is important for the Cressman interpolation below. Can you show a map of the stations?*

**Response:** There are 3087 stations in the region from15-55N and 70-140E. We have showed the stations in Figure 1.

*L94: how were these thresholds decided? How do they compare to climatology?*

**Response:** At present, there are two kinds of visibility thresholds for haze in the world. The value recommended by WMO is no more than 5km. Before 1948 in China, the maximum visibility of all kinds of horizontal obstruction to vision affecting visibility was less than 4 km. Clear air is a phenomenon of good atmospheric transparency (visibility >= 10km). After that, the maximum visibility of haze increased to less than 10km, which is still used up to now. According to WMO's 1984 report, the relative humidity of haze is less than 80%. These thresholds are constantly updated according to actual needs, which is good compare to climatology. For example, Liu et al. (2017) and Zhao et al. (2017) used these thresholds.

[Figure]

Figure 1: The conceptual model of distinguishing haze and mist or fog (Wu et al., 2018)

*L98: Is Cressman objective analysis the same as Cressman interpolation method in the last sentence? please clarify*

*L104: how are these values chosen? Which value out of 5, 4, 3 is used for which variable? Do you show results for all three values? This needs more clarity. I could not reproduce your methods based on the level of detail given here.*

**Response:** Yes, Cressman objective analysis is the same as Cressman interpolation method. The sentences have been revised. Cressman objective analysis is a successive corrections process using multiple Cressman interpolations.

The value of R is no more than 10, generally 1-4. The value decreases in turn, indicating the radius

of each successive corrections. There is always a subjective factor for choosing these values, generally 10, 7, 4, 2, 1. In this paper, R is 5, 4 and 3, which is more than 2 and less than 7. If Cressman objective analysis is not used, the figure is as follows, which is similar to the figure 2 in our manuscript. Compared with the figure 2a (without interpolation), the shading in figure 2 (interpolation) in our manuscript becomes smoother.

[Figure]

Figure 2a: Spatial distribution of monthly mean visibility (unit: km, shading) (without interpolation) in (a) November and (b) December 2015. The black box indicates the NCP (30°N-40.5°N, 112°E-121.5°E). Shading indicates the value of visibility. The black dots represent the location of meteorological stations.

*L109: In the formulation of Plumb (1985) the primes in this equation are deviations from the zonal mean not the time mean state. Below in equation 10 you describe calculating the prime by taking the deviation from the time mean state. Can you clarify?*

**Response:** Yes, the perturbation stream function is calculated according to the zonal mean flow. We mean that the geopotential height anomaly used in the calculation of the perturbation stream function is the mean relative to the climate state. The equation 10 we describe mainly refers to the calculation of other anomalies mentioned in the manuscripts. For example, figure 4 is based on the equation 10.

*L176: Does the cold air induce anomalous vorticity that generates the wave? Or was the anomalous vorticity that forces the wave train already present? please clarify*

**Response:** Yes, it is cold air that induce anomalous vorticity that generates the wave. At upper troposphere, there was a strong northerly anomaly over the European continent. The strong northerly wind intruded into the entrance of the subtropical westerly jet over the vicinity of the middle eastern Mediterranean and formed the cold air convergence (Ding and Li, 2017) (Figure 4a, Figure 5a), named as western disturbance (Syed et al., 2006). Within the subtropical westerly jet, there existed a southerly/northerly wind wave train (Li, 1988; Li and Sun, 2015; Ding and Li, 2017).

Li et al. (2015) found the Rossby wave originated from strong cold air intrusion into the subtropical westerly jet over the eastern Mediterranean. Another feature was cold advection with northerly wind over northeastern Africa (eastern Asia), coupled with upper tropospheric convergences and mid-troposphere southwest–northeast (west–east) troughs, which induced the propagation of Rossby

energy southeastward from the midlatitudes to the North African-Asian jet (Li, 1988).

*L316: no I dont think it does show this directly. Instead the correlation shows that the EUI you have defined captures similar variance in the meridional wind field to the EOF.*

**Response:** The results of EUI and empirical orthogonal function (EOF) of the 200 hPa meridional winds represent different wave trains, and their positions are obviously different. The EUI defined in our manuscripts is similar to the previous EU index, which represents the North Branch wave train (the Eurasian teleconnection), while the empirical orthogonal function (EOF) of the 200 hPa meridional winds mainly represents the South Branch wave train. There is a high correlation coefficient between EUI defined by us and the meridional wind field to the EOF, indicating that both exist when haze occurs, which is consistent with the previous results mentioned in the manuscripts.

**References**

China Meteorological Administration.: Introduction to the standard for observation, identification and classification of haze (in Chinese), 6-7, Press, Beijing, China, 2014.

Ding, F. and Li, C.: Subtropical westerly jet waveguide and winter persistent heavy rainfall in south China, J. Geophys. Res. Atmos., 122, 7385-7400, doi:10.1002/2017JD026530, 2017.

Li, C. and Sun, J. L.: Role of the subtropical westerly jet waveguide in a southern China heavy rainstorm in December 2013, Adv. Atmos. Sci., 32, 601-612, doi:10.1007/s00376-014-4099-y, 2015.

Li, C. Y.: Frequent activity of East Asian trough and El Nino onset, Sci. China Ser. (B), 31, 667–674, 1988. (in Chinese)

Liu, Q., Sheng, L., Cao, Z., Diao, Y., Wang, W., and Zhou, Y.: Dual effects of the winter monsoon on haze-fog variations in eastern China, J. Geophys. Res. Atmos., 122, 5857-5869, doi:10.1002/2016JD026296, 2017.

Syed, F. S., Giorgi, F., Pal, J. S., and King, M. P.: Effect of remote forcings on the winter precipitation of central southwest Asia Part 1: Observations, Theor. Appl. Climatol., 86, 147-160, doi:10.1007/s00704-005-0217-1, 2006.

Wu, Y. X., Wu, D.: Historical evolution and revised proposals of observation standards (norms) of haze, Environmental Science & Technology, 41 (10): 206-212, 2018.

Zhao, S., Li, J. P., and Cheng S.: Decadal variability in the occurrence of wintertime haze in central eastern China tied to the Pacific Decadal Oscillation, Scientific Reports, 6:27424, doi: 10.1038/srep27424, 2017.

---

## Editor Decision (ED2)

[revised manuscript text omitted]

**Figure 8:** The 200 hPa horizontal wind anomalies (vector, $\text{m s}^{-1}$) and meridional wind anomalies (contours, $\text{m s}^{-1}$) regressed onto the standardized -PC1 of the leading EOF mode for 200 hPa meridional wind anomalies (a) and the standardized PC1 of the leading EOF mode for 200 hPa zonal wind anomalies (b). The values of the contours and arrows are the regression coefficients. The solid red (dashed blue) contours indicate the positive (negative) meridional wind anomaly. The thick black line in (a) and (b) delineates the climatological jet axis (the gradient of the zonal wind with the longitude is 0). Only anomalies statistically significant at the 0.05 level based on a two-sided Student's t test are shown in (a) and (b).

[Figure]

**Figure 9: Regression of 500 hPa geopotential height anomalies and 850 hPa wind vector onto the negative standardized**

530   **-PC1 of the leading EOF mode for the 200 hPa meridional wind anomaly. The values of shading (units, gpm) and arrows (units, m/s) represent regression coefficients. Regions of dotted areas indicate anomalies exceeding the 0.05 confidence level. Only anomalies statistically significant at the 0.05 level based on Student's t test are given for the wind vector.**

[Figure]

**Figure 10: Anomalous geopotential height (shading, 10 gpm) at 500 hPa in November and December 2015 and its stationary wave activity flux (vector, $\mathrm{m^{-2}\ s^{-2}}$).**

[Figure]

**Figure 11: A schematic diagram of the negative EU in the 200 hPa geopotential height field and meridional wind anomaly at 500 hPa in November and December 2015. H (L) denotes positive (negative) geopotential height anomalies; the plus and minus represent positive (southerly) and negative (northerly) meridional wind anomalies, respectively; and the shaded belt of arrow represents the subtropical westerly jet.**

[Figure]

**Figure 12: Schematic illustration showing circulation system affecting the haze events in November and December 2015. The red (blue) circular of arrow represents convergence (divergence); the red (blue) translucent arrow represents ascending (descending) air; the brown translucent arrow represents the southerly wind anomaly; the purple solid arrow represents the water vapor transported by southwesterly airflow from the Bay of Bengal and South China Sea; the thick (thin) black arrow represents northward (southward) movement in the atmosphere; H denotes positive geopotential height anomaly; the white weather symbol represents precipitation.**

[Figure]

**Figure 13: The mean visibility anomaly (dashed line, purple) in November and December in the NCP, the EUI (solid line, green), and PC1 (bars, red and blue) of the leading EOF mode for the 200 hPa meridional wind anomaly from 1980 to 2015. The linear regression coefficient of the mean visibility onto PC1 is approximately 0.38 km, which is statistically significant at the 95% confidence level.**

---

## Author Response (AR3)

Dear editor,

We appreciate you for carefully rreviewing our manuscripts and providing the suggestion again to improve the quality of it. We have carefully read your comments and revised the manuscripts as suggested. We marked all relevant changes in the manuscript in red.